# Structural and functional analysis of the role of the chaperonin CCT in mTOR complex assembly

Jorge Cuéllar[1], W. Grant Ludlam [2], Nicole C. Tensmeyer [2], Takuma Aoba[2], Madhura Dhavale[2], César Santiago[1], M. Teresa Bueno-Carrasco[1], Michael J. Mann[2], Rebecca L. Plimpton[2], Aman Makaju[3], Sarah Franklin[3], Barry M. Willardson[2] & José M. Valpuesta [1]

The mechanistic target of rapamycin (mTOR) kinase forms two multi-protein signaling complexes, mTORC1 and mTORC2, which are master regulators of cell growth, metabolism, survival and autophagy. Two of the subunits of these complexes are mLST8 and Raptor, β-propeller proteins that stabilize the mTOR kinase and recruit substrates, respectively. Here we report that the eukaryotic chaperonin CCT plays a key role in mTORC assembly and signaling by folding both mLST8 and Raptor. A high resolution (4.0 Å) cryo-EM structure of the human mLST8-CCT intermediate isolated directly from cells shows mLST8 in a near-native state bound to CCT deep within the folding chamber between the two CCT rings, and interacting mainly with the disordered N- and C-termini of specific CCT subunits of both rings. These findings describe a unique function of CCT in mTORC assembly and a distinct binding site in CCT for mLST8, far from those found for similar β-propeller proteins.

[1] Centro Nacional de Biotecnología, Campus de la Universidad Autónoma de Madrid, 28049 Madrid, Spain. [2] Department of Chemistry and Biochemistry, Brigham Young University, Provo, UT 84602, USA. [3] Department of Internal Medicine, Nora Eccles Harrison Cardiovascular Research and Training Institute, University of Utah, Salt Lake City, UT 84112, USA. Correspondence and requests for materials should be addressed to B.M.W. (email: bmwillardson@chem.byu.edu) or to J.M.V. (email: jmv@cnb.csic.es)

The mechanistic target of rapamycin (mTOR) protein kinase is a master regulator of cell growth, metabolism, and survival, and as such, it constitutes a high-value drug target[1]. mTOR interacts with mammalian lethal with SEC13 protein 8 (mLST8) and regulatory associated protein of mTOR (Raptor) to form mTOR complex 1 (mTORC1)[2], or with mLST8, rapamycin-insensitive companion of mTOR (Rictor), and mammalian stress-activated MAP kinase-interacting protein 1 (mSIN1) to form mTOR complex 2 (mTORC2)[3]. These complexes are functionally distinct as mTORC1 is activated by growth factors and amino acids to promote protein, lipid, and nucleic acid synthesis and inhibit autophagy, while mTORC2 functions upstream of mTORC1 in growth factor signaling to activate cell survival pathways by phosphorylating the kinases AKT, PKC, and SGK1[1].

In order to perform their signaling functions, the mTOR complexes must be assembled from their nascent polypeptides. Protein complex assembly is often mediated by molecular chaperones that assist nascent or misfolded proteins to achieve their native structures and assemble into functional complexes[4]. Protein folding and complex formation seldom occurs spontaneously in the very concentrated protein environment of the cell, but requires chaperones to protect proteins from aggregation, to channel their folding pathways and to facilitate their association into multiprotein assemblies[4]. The mTOR kinase is a 289 kDa protein that requires the Hsp90 chaperone and the Tel2-Tti1-Tti2 (TTT)-R2TP co-chaperone complex to fold properly[5,6]. However, little is known about how the other mTORC components are folded and brought together with mTOR. Yeast genetic studies have pointed to a possible role for the cytosolic chaperonin containing TCP-1 (CCT, also called TRiC) in mTOR complex formation. Overexpression of CCT subunits suppressed phenotypes associated with temperature sensitive mutations of yeast TOR and LST8, indicating a genetic interaction between CCT and the yeast TOR complex[7,8]. Furthermore, genetic disruption of CCT ATPase activity resulted in phenotypes similar to those observed with loss of yeast TOR signaling[9]. These findings in yeast are consistent with results from human interactome studies that identified interactions between mLST8 and Raptor with CCT, but not with the other mTORC components[10].

CCT is a eukaryotic member of the chaperonins, which are divided in two types: type I, which is present in eubacteria and in organelles of endosymbiotic origin; and type II, which is present in archaea and the eukaryotic cytosol. All are large oligomers that form a double-ring structure[11,12]. CCT is the most complex of all chaperonins with each of the two rings composed of eight paralogous subunits (referred to here as CCT 1–8). At the center of each ring is a protein folding chamber measuring approximately 60 Å in diameter[13] with a volume large enough to encapsulate a 70 kDa protein[14]. Each of the subunits of CCT and the other chaperonins can be divided into three domains: the equatorial domain, which hosts the ATP-binding site and most of the intra- and inter-ring interactions; the apical domain, which is believed to be responsible for substrate recognition and binding; and the intermediate domain, which acts as a linker between the other two domains. ATP binding and hydrolysis in the CCT subunits induce conformational changes in the CCT structure that drive protein folding[15,16]. Unfolded polypeptides bind within the folding chamber when CCT is in its open conformation and the nucleotide-binding sites are empty[17,18]. As ATP binds and the ATP hydrolysis transition state is achieved, the chamber closes, due to the movement of a long α-helical protrusion in each subunit, and the protein is trapped within the chamber[19–21]. This entrapment assists folding by confining the degrees of conformational freedom of the polypeptide and by influencing the folding trajectory[11,12]. After ATP hydrolysis, the chamber opens and if the protein has achieved a native fold and lost its contacts within the chamber, it is released.

CCT assists in folding proteins with multiple domains or complex folds and helps to assemble multi-protein complexes[18,22]. Among these, proteins with β-propeller domains are an important class of CCT folding substrates[23,24]. β-propeller domains commonly consist of seven WD40 repeat sequences that fold into seven β-sheets that form the blades of a propeller-like circular structure[25]. β-propellers have a unique folding trajectory that requires the C-terminus to interact with the N-terminus to make the last β-sheet that closes the β-propeller. CCT may help bring the termini together and assist the β-propeller to close during folding[23]. These β-propeller domains have important functional roles from protein-protein interactions to enzymatic catalysis. β-propeller proteins that are folded by CCT include G protein β subunits (Gβ)[18,26], the cdc20 and cdh1 components of the anaphase promoting complex[24], and the protein phosphatase 2A regulatory subunits[27] among others.

Two of the subunits of mTOR complexes, mLST8 and Raptor, contain β-propeller domains[2,28]. mLST8 consists entirely of a single β-propeller that binds and stabilizes the mTOR kinase domain[2,28], while Raptor contains a C-terminal β-propeller[2,29] that may bind regulatory proteins[2,30]. The mLST8 β-propeller shows strong structural homology with the β-propeller of Gβ[25,28], further suggesting that mLST8 may be folded by CCT. To test this possibility, we used functional and structural approaches to investigate the role of CCT in mTORC formation and signaling. Our findings suggest that CCT contributes to mTORC assembly and signaling by folding the mLST8 and Raptor β-propellers. We solved the structure of the mLST8–CCT intermediate in mTORC assembly by cryo-EM to 4.0 Å. At this resolution, the structure shows an almost native mLST8 β-propeller bound to CCT in an unexpected position deep within the folding chamber between the two CCT rings, revealing a unique means by which a β-propeller substrate is recognized by CCT.

## Results

**mLST8 and Raptor β-propellers bind CCT.** To begin to examine the possible role of CCT in mTORC assembly and function, we sought to confirm the interaction of mLST8 and Raptor with CCT reported in human interactome studies[10]. We ectopically expressed human mLST8 or Raptor in cells and assessed their binding to endogenous CCT by co-immunoprecipitation. With mLST8, we observed strong co-immunoprecipitation when either mLST8 or CCT5 was immunoprecipitated, indicating a robust interaction between mLST8 and CCT (Fig. 1a, b). Similar results were observed with human Raptor and CCT5 (Fig. 1c, d). To determine the domain of Raptor responsible for the interaction, we expressed Raptor truncations containing the C-terminal β-propeller (residues 1000–1335) or the N-terminal caspase homology and armadillo repeat domains (residues 1–999) and measured their binding to CCT by co-immunoprecipitation (Fig. 1e, f). The C-terminal β-propeller domain bound CCT robustly while the N-terminal domains showed no interaction, indicating that Raptor binds CCT through its β-propeller domain. We also tested the binding of mTOR and the other core components of mTORC2, Rictor, and mSIN1 to CCT by co-immunoprecipitation and found no interaction (Supplementary Fig. 1). Collectively, these findings demonstrate interactions of the mLST8 and Raptor β-propellers with CCT and suggest that CCT might be involved in their folding.

To further test the possibility that the mLST8 and Raptor β-propellers are folded by CCT, we measured the effect of ATP on their co-immunoprecipitation with CCT. Substrates are known to release from CCT in an ATP-dependent manner[31]. We incubated

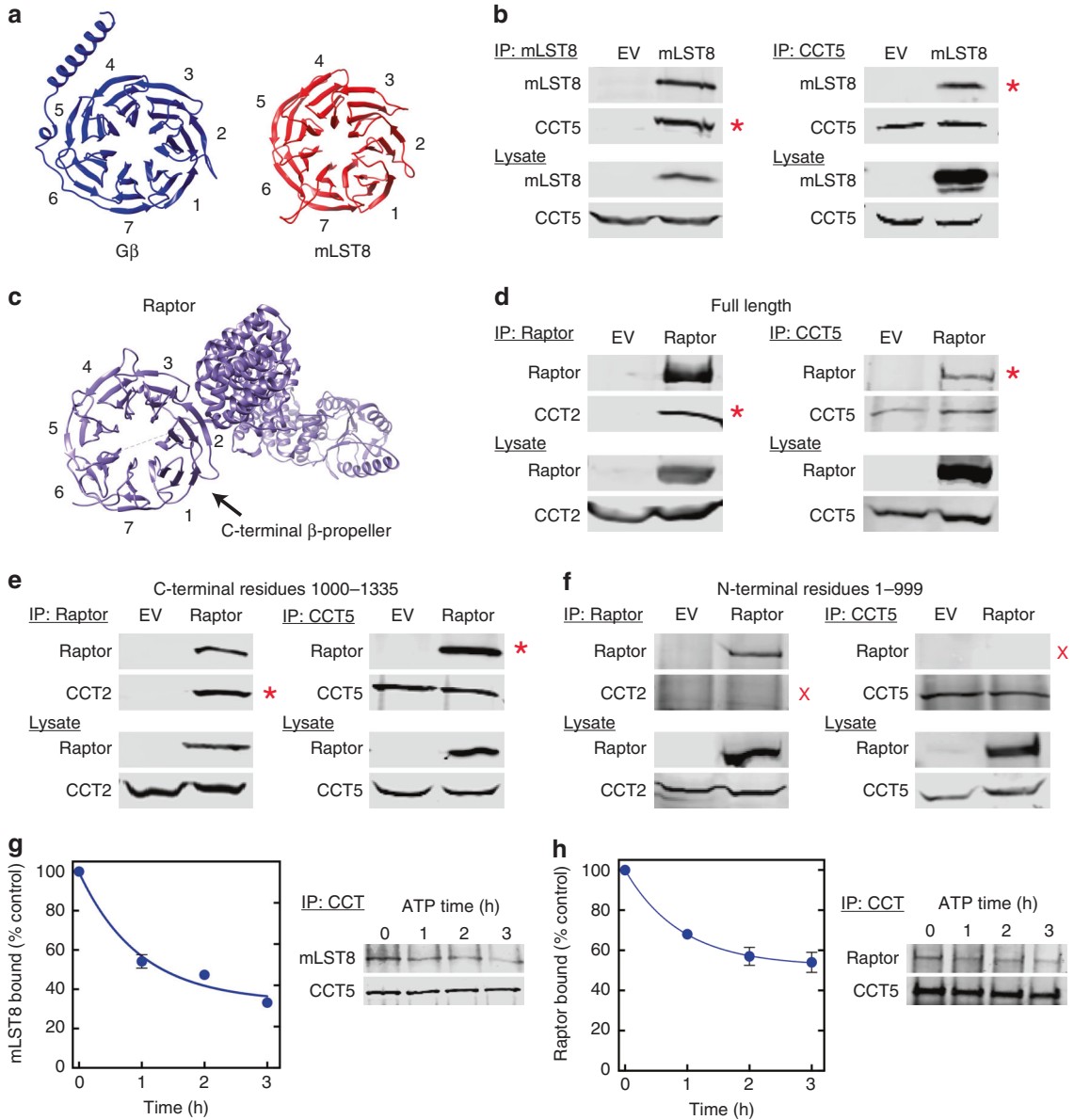

**Fig. 1** mLST8 and Raptor bind CCT. **a** β-propeller structures of Gβ (PDB 1TBG) and mLST8 (PDB 4JT6). The β-sheets are numbered according to convention with blade 7 containing β-strands from both the N- and C-termini. **b** Co-immunoprecipitation of mLST8 and CCT. HEK-293T cells were transfected with mLST8 or an empty vector (EV), immunoprecipitated, and immunoblotted as indicated. Co-immunoprecipitating bands are marked (red asterisks). **c** Raptor structure (PDB 5EF5) with β-sheets numbered as in panel (**a**). **d-f** Co-immunoprecipitation of Raptor and CCT. Cells were transfected with full length HA-tagged Raptor (**d**), or HA-tagged constructs containing the C-terminal β-propeller (**e**) or the N-terminal caspase and armadillo domains (**f**). Control cells were transfected with empty vector as indicated. Cells were immunoprecipitated and immunoblotted as indicated. All blots are representative of at least three separate experiments. **g** ATP causes release of mLST8 from CCT. CCT immunoprecipitates from cells overexpressing mLST8 were treated with 5 mM ATP for the times indicated, washed and immunoblotted for mLST8 and CCT5. The amount of mLST8 remaining is shown as a percent of the no ATP control. Error bars smaller than the symbols are not visible. **h** ATP causes release of Raptor from CCT. The CCT immunoprecipitation experiment was repeated with cells overexpressing Raptor. Source data are provided in the Source Data file

our CCT immunoprecipitates containing mLST8 (Fig. 1g) or Raptor (Fig. 1h) with 5 mM ATP and measured the amount of each that remained bound over time. ATP decreased mLST8 and Raptor binding to CCT in a time-dependent manner, reaching a steady-state reduction of 60% with a half-life of 40 min for mLST8 and a steady-state reduction of 50% with a half-life of 37 min for Raptor, indicating that both are indeed CCT folding substrates. These release rates are significantly slower than the 7 min half-life reported for release of actin from CCT under different in vitro conditions[32], suggesting that slow release may be a common property of WD40 proteins.

**CCT contributes to mTORC assembly and signaling**. To assess the contribution of CCT to mTOR complex formation, we sought a method to genetically deplete CCT from cells without compromising viability. The CCT complex is essential and cannot be deleted without causing cell death over time. To resolve this issue, we chose a CRISPR approach that decreased expression of CCT in a cell population significantly without completely eliminating it (Fig. 2a, see Methods). In these cells, CCT expression was reduced by 80%, resulting in significant decreases in expression of endogenous mTOR (65%), mLST8 (40%), and Raptor (50%), but not Rictor, mSIN1, or the

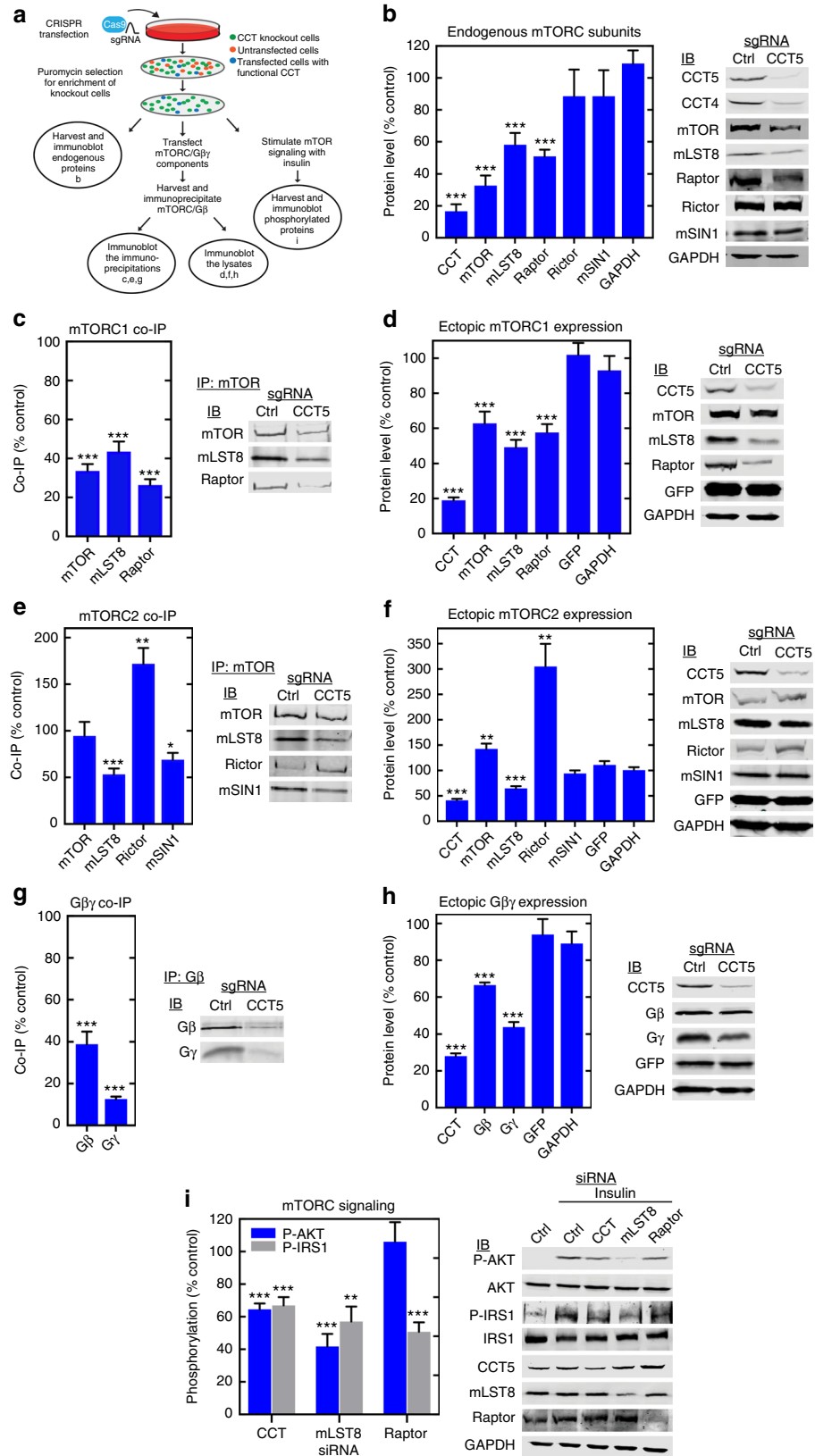

GAPDH control (Fig. 2b), indicating that the effect was specific. The changes in expression occurred post-transcriptionally because CCT depletion had no effect on mTOR, mLST8, or Raptor mRNA levels (Supplementary Fig. 2a). Thus, the decreases in mLST8 and Raptor expression are most likely due to their inability to fold, while the decrease in mTOR expression

may result from an inability to form stable complexes in the absence of CCT.

To examine further the contribution of CCT in mTORC1 assembly, we ectopically expressed mTOR, Raptor, and mLST8 in CCT-depleted cells and assessed mTORC1 formation by measuring the co-immunoprecipitation of Raptor and mLST8

**Fig. 2** CCT contributes to mTORC assembly. **a** Work flow of the CCT depletion experiments. Bold letters correspond to the figure panels in which the experimental results are displayed. **b** Effects of CCT depletion on endogenous mTORC subunit expression. Cells were treated with CCT5 sgRNA or control sgRNA and Cas9, lysates were immunoblotted and band intensities were quantified as indicated. Data are shown as a percent of the control. Bars represent the average ± standard error. **c** Effects of CCT depletion on co-immunoprecipitation of mTORC1 subunits. Cells were CRISPR treated and transfected with the indicated mTORC1 subunits and a GFP control. mTOR immunoprecipitates were immunoblotted and quantified as indicated. **d** Lysates from cells in panel (**c**) were immunoblotted and quantified for expression of mTORC1 subunits and controls as indicated. **e** Effects of CCT depletion on co-immunoprecipitation of mTORC2 subunits with mTOR. Cells were CRISPR treated and transfected with the indicated mTORC2 components and a GFP control. mTOR immunoprecipitates were immunoblotted and quantified as indicated. **f** Lysates from the cells in panel (**e**) were immunoblotted and quantified for expression of mTORC2 subunits and controls as indicated. **g** Effects of CCT depletion on co-immunoprecipitation of Gγ with Gβ. Cells were CRISPR treated and transfected with Gβ, Gγ, and a GFP control. Gβ immunoprecipitates were immunoblotted and quantified as indicated. **h** Lysates from the cells in panel g were immunoblotted and quantified for expression of Gβ and Gγ and controls as indicated. **i** Effects of siRNA-mediated CCT depletion on insulin-mediated IRS1 S636/639 phosphorylation or AKT S473 phosphorylation in HEPG2 cells. Cells were treated with siRNAs to CCT1/CCT5, mLST8, Raptor or a non-targeting control, serum starved for 18 h and then treated with insulin. Cell lysates were immunoblotted as indicated. *$p < 0.05$, **$p < 0.01$, ***$p < 0.005$. Source data are provided in the Source Data file

with mTOR as well as the cellular expression of each subunit. CCT depletion resulted in a 65% decrease in mTOR immuno-precipitation with corresponding decreases in mLST8 and Raptor co-immunoprecipitation (Fig. 2c). This decrease in mTORC1 formation could be attributed to similar decreases in mTOR, mLST8, and Raptor expression upon CCT depletion (Fig. 2d). The effect appeared specific because ectopic expression of GFP or endogenous expression of GAPDH was unchanged with the loss of CCT (Fig. 2d). The decrease in mTORC1 components upon CCT depletion is similar to that seen with Gβ, a known CCT folding substrate (Fig. 2g, h), but less than Gγ, a small 70 amino-acid protein that is rapidly degraded when the Gβγ dimer cannot form[26,33]. Collectively, these results suggest that CCT contributes significantly to mTORC1 formation. When CCT is depleted, mLST8 and Raptor are destabilized, which results in less mTORC1 assembly and decreased mTOR expression.

In the case of mTORC2, CCT depletion did not change mTOR immunoprecipitation, but it caused a decrease in mLST8 (50%) and mSIN1 (30%) co-immunoprecipitation (Fig. 2e). Unexpectedly, Rictor co-immunoprecipitation increased by 70%. These changes generally paralleled the changes in ectopic expression of the subunits, which showed an increase in mTOR (40%), a striking increase in Rictor (300%), a decrease in mLST8 (50%) and no change in mSIN1 (Fig. 2f). The effects appeared specific because expression of the controls was unchanged. These findings suggest that mLST8 incorporation into mTORC2 also depends on CCT, but that CCT depletion significantly increases Rictor expression and association with mTOR, perhaps as a result of decreased endogenous Raptor, given that Rictor and Raptor are known to compete for mTOR binding[34]. To test this possibility, we measured the effects of CRISPR-mediated Raptor depletion on ectopic expression of mTORC2 subunits. Raptor loss resulted in an increase in Rictor expression, while there was little change in the other mTORC2 subunits (Supplementary Fig. 2b). These results suggest that CCT depletion decreases Raptor expression, which causes a compensatory increase in Rictor expression under these conditions.

CCT contributions to mTORC assembly should also be reflected in mTOR signaling. To test this possibility, we assessed the effects of CCT depletion on mTOR-dependent phosphorylation downstream of insulin in HEPG2 cells. Cells were siRNA-depleted of CCT and treated with insulin. mTORC1 activity was assessed by IRS1 S636/S639 phosphorylation[35], while mTORC2 activity was assessed by AKT S473 phosphorylation[36]. For comparison, cells were also siRNA-depleted of mLST8 or Raptor. CCT depletion resulted in a 40% inhibition of both IRS1 and AKT phosphorylation, which was similar to the decreases observed with mLST8 depletion (Fig. 2i). Likewise, Raptor depletion caused a 50% decrease in IRS1 phosphorylation, but

showed no change in AKT phosphorylation as expected (AKT is only a substrate of mTORC2). These results support the idea that CCT participates in mTOR signaling by assisting in the incorporation of mLST8 and Raptor into mTORC1 and mLST8 into mTORC2.

**PhLP1 does not assist in mTORC assembly**. The participation of CCT in mTOR complex formation raises the possibility that the CCT co-chaperone PhLP1 may also be involved, especially since PhLP1 is required for Gβ to fold and assemble into the Gβγ dimer[33]. To test this possibility, we used the same CRISPR strategy to deplete cells of PhLP1 and measure the effects on mTOR subunit expression and assembly. Unexpectedly, there was no decrease in expression of endogenous mTORC subunits despite a 70% reduction in PhLP1 (Supplementary Fig. 3a). Likewise, there was no change in formation of mTORC1 and mTORC2 (Supplementary Fig. 3b–e). In contrast, this same PhLP1 depletion resulted in a striking 90% decrease in Gγ association with Gβ as expected (Supplementary Fig. 3f, g). These results argue against a contribution of PhLP1 to mLST8 or Raptor folding and highlight differences between Gβ and mLST8 folding despite their structural similarities.

**Cryo-EM structure of the mLST8–CCT assembly intermediate**. To investigate the mechanism of mLST8 and Raptor folding by CCT, we sought to isolate CCT-bound intermediates directly from cells and characterize their structures using cryo-electron microscopy (cryo-EM) and single-particle 3D reconstruction. We were successful at isolating human mLST8 bound to endogenous CCT from HEK-293T cells overexpressing mLST8 using a tandem affinity chromatography strategy without adding exogenous nucleotide (Supplementary Fig. 4a). However, Raptor-CCT complexes could not be readily isolated using a similar strategy, possibly due to its large N-terminal domain which does not interact with CCT and might result in a less stable complex. Therefore, we focused our structural characterization efforts on the mLST8–CCT complex.

The purified mLST8–CCT complex was vitrified on grids and images were recorded with a Titan Krios electron microscope (Supplementary Fig. 4b). A total of 1,769,600 particles were selected and subjected to 2D classification (Supplementary Fig. 4c and Supplementary Table 1). The classes showed mostly the two typical views, the end-on and side orientations, and the best classes (1,197,358 particles) were subjected to a set of 3D classifications from which 452,000 particles were selected for further processing. The angular coverage was very good (Supplementary Fig. 4d) and the final 3D reconstruction reached 4.0 Å resolution after postprocessing (Supplementary Fig. 4e).

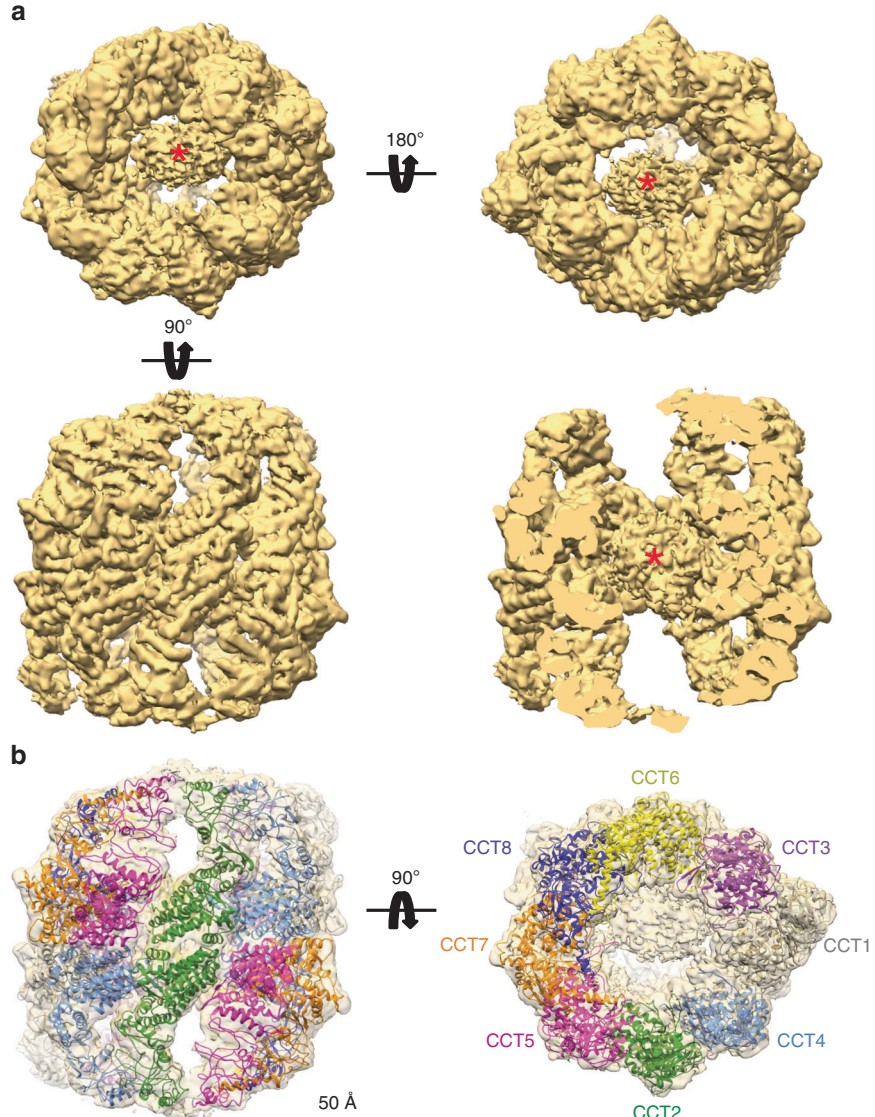

**Fig. 3** Cryo-EM structure of the mLST8–CCT complex. **a** Top, the two end-on views of the 3D reconstruction of the mLST8–CCT complex. Bottom, side view of the complex, either intact (left) or sliced through the center of the mass (right), to show the presence of the mLST8 molecule (red asterisks). **b** Docking of the human CCT atomic model into mLST8–CCT 3D reconstruction. The color scheme for the CCT subunits is kept the same in all the figures. Bar indicates 50 Å

The resolution was not isotropic with the central, equatorial domains yielding higher resolution than the flexible apical domains (Supplementary Fig. 4f, g). The reconstruction shows human CCT in an open conformation with a prominent mass located in the center of the structure between the two CCT rings (Fig. 3a). The apical domains have a very asymmetric arrangement in which the eight subunits in each ring are arranged as a tetramer of dimers as observed previously for yeast CCT[13]. Besides the large central mass, the structure is similar to those of bovine and yeast CCT in their open conformations[13,17,32], in particular to the structure described by Zang et al.[13] for yeast CCT in the AMP–PNP-bound conformation, despite the fact that the mLST8–CCT was purified in the absence of added nucleotide.

Docking of the atomic model of yeast CCT in the AMP–PNP conformation[13] (PDB 5GW5) into the mLST8–CCT reconstruction confirms the similarity between the two structures (Supplementary Fig. 5a). The docking is almost perfect in the equatorial domains and very good in the intermediate and the base of the apical domains, which allowed us to unambiguously assign the different subunits of the chaperonin in the mLST8–CCT complex (Supplementary Fig. 5a). The most important differences in the docking are located in the helical protrusions, which are angled downward more toward the center of the central cavity in the mLST8–CCT structure (Supplementary Fig. 6). We used the atomic model of the yeast CCT in the AMP–PNP conformation and the sequences of the human CCT subunits to generate an atomic model that we subjected to flexible docking into the 3D reconstruction of the mLST8–CCT complex using the program IMODfit[37]. This atomic model was subsequently refined applying the real-space refinement protocol in PHENIX[38] and Refmac5 in CCPEM. Regions where density was absent were eliminated from the final model. This structure provides a high-resolution atomic model for human CCT (Fig. 3b, Supplementary Fig. 5b and Supplementary Table 2).

**Position of mLST8 in the CCT folding cavity.** The most striking difference between the human mLST8–CCT structure and CCT structures from bovine and yeast[13,17,32] is the presence of the

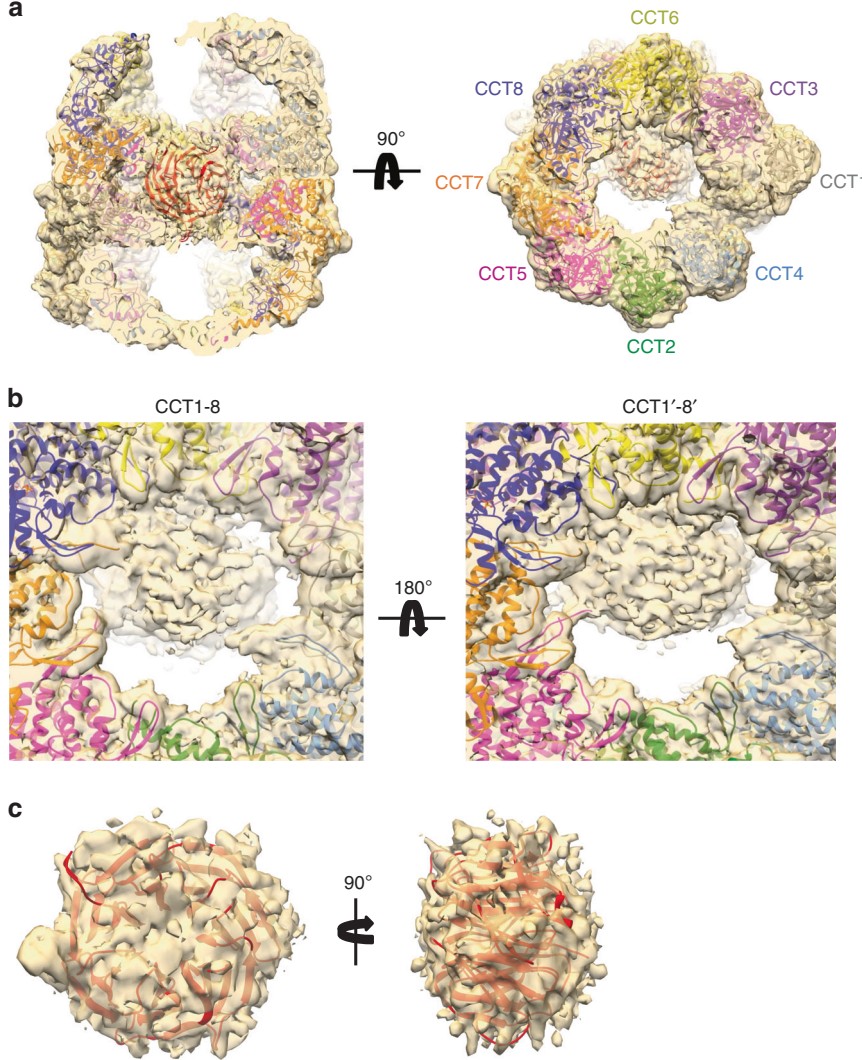

**Fig. 4** Position of mLST8 within the mLST8–CCT complex. **a** Docking of the atomic structure of mLST8 (red, PDB 4JT6) into the corresponding mass of the 3D reconstruction of the CCT–mLST8 complex. Left, sliced side view, and right, end-on view, show how mLST8 fits in the corresponding density. **b** Detailed views of the CCT subunits of each ring involved in mLST8 interaction, showing how the unstructured N- and C-termini of several subunits contact mLST8. Left, top ring, shows interactions with CCT5, CCT7, CCT8, CCT6, and CCT3, and right, bottom ring, with CCT5′, CCT7′, CCT8′, CCT6′, and CCT1′. **c** The extracted cryo-EM density from between the rings with mLST8 docked shows the quality of the fit of mLST8 into this density

mass in the interior of the cavity, positioned between the two CCT rings and contacting both rings (asterisks in Fig. 3). The mass has a circular shape, indicative of a β-propeller, and docking of the atomic structure of mLST8, as found in the mTORC1 complex[28], is very good (Fig. 4, Supplementary Movie 1). The quality of the fit can be clearly seen when the mass attributable to mLST8 along with the docked mLST8 atomic structure is extracted from the structure (Fig. 4c). These observations suggest that the mass corresponds to mLST8 in a stable conformation that resembles the native state. However, some internal mass has been detected at the level of the equatorial domains in previous 3D reconstructions of substrate-free CCT[13,39]. Thus, to determine if the mass we observed was indeed mLST8, we carried out a 3D reconstruction of human CCT in the absence of mLST8 and compared the structures. For this, we isolated substrate-free human CCT from HEK-293T cells without mLST8 over-expression (Supplementary Fig. 7a). The purified CCT was vitrified on grids and images recorded in a Titan Krios. A total of 504,060 particles were selected and subjected to a 2D classification. The best classes (139,819 particles) were used for a 3D reconstruction, which attained 7.5 Å resolution (Supplementary

Fig. 7b and Supplementary Table 1). The resulting volume, albeit at a lower resolution, shows similar structural features to that of the mLST8–CCT complex, with an open and asymmetrical distribution in the apical domains. The major difference between the two 3D reconstructions is the extent and shape of the mass resolved in the cavities of the CCT structure (Supplementary Fig. 7c). The mass in the mLST8–CCT complex is large and accounts for the β-propeller structure of mLST8, while that of the substrate-free CCT is much smaller and can only be explained as part of the N- and C-terminal disordered regions of the CCT subunits known to reside at the bottom of the folding cavity[19,40]. This result reinforces the assignment of the internal mass between the rings to mLST8 in the mLST8–CCT structure.

In addition to its location between the CCT rings, mLST8 is situated on one side of the central cavity near the CCT3, 6, and 8 subunits and interacts with the disordered regions belonging to the N- and C-terminus of several CCT subunits (Fig. 4b). These interactions involve the termini of CCT5, CCT7, CCT8, CCT6, and CCT3 in one of the rings, and CCT5′, CCT7′, CCT8′, CCT6′, and to a lesser extent CCT1′ in the other ring (Fig. 4b). Several studies have revealed the presence of two functional hemispheres

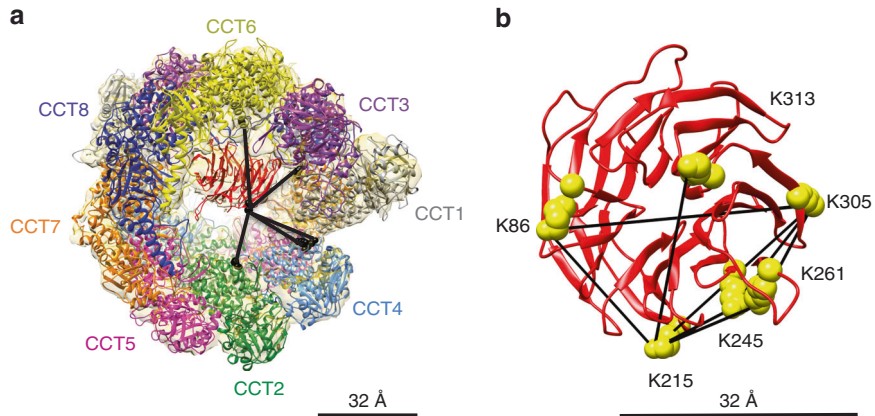

**Fig. 5** XL-MS of the mLST8–CCT complex. **a** Intermolecular cross-links used to orient mLST8 in the cryo-EM density are mapped onto the mLST8–CCT structure. The last ordered residue of the termini containing the lysine involved in cross-links are shown as black spheres. **b** Intramolecular mLST8 cross-links are mapped onto the native structure of mLST8 (PDB 4JT6). The XL-MS pLink output is provided in the Source Data file

in the CCT oligomer[13,40–42], one formed by the adjacent CCT5, CCT2, CCT4, and CCT1 subunits on one side (the CCT2 hemisphere) and the other formed by CCT3, CCT6, CCT8 and CCT7 on the opposite side (the CCT6 hemisphere). The CCT2 hemisphere shows strong ATP binding and hydrolysis while the CCT6 hemisphere shows much weaker ATP binding and hydrolysis[43]. Collectively, these structural observations indicate that the mLST8 molecule is bound between the two chaperonin rings on the low-ATP binding side of the rings (the CCT6 hemisphere) through interactions with the disordered termini of these subunits.

**Cross-linking mass spectrometry**. To further assess the location of mLST8 in the CCT folding cavity, we turned to chemical cross-linking coupled with mass spectrometry (XL-MS), which provides distance constraints that confirm the position of subunits within protein complexes. We treated the mLST8–CCT complex with disuccinimyl suberate (DSS), which cross-links adjacent lysine residues with a maximal distance of ~ 32 Å between their Cα carbons, taking into account the length of the lysine side chains and typical peptide backbone flexibility[44]. The cross-linked complex was protease-digested and the resulting peptides analyzed by MS. The MS data were searched for cross-linked peptides using the pLink2 search engine[45]. The analysis detected 196 cross-links within and between CCT subunits, 5 cross-links between mLST8 and CCT subunits, and 8 cross-links within mLST8 itself (Fig. 5, Supplementary Table 3). We compared the 48 cross-links within the conformationally stable equatorial domains to the structural model and found that all fit the distance constraints (Supplementary Table 3), supporting the accuracy of the structural model and the quality of the cross-linking data. The five intermolecular links involved K215 of mLST8 cross-linked to lysines in the disordered regions of the N- and C-termini of adjacent CCT subunits (Fig. 5a). The termini are located at the bottom of the CCT folding cavity near the interface between the CCT rings, extending toward the center of the rings. These cross-links are consistent with the position of mLST8 between the CCT rings and support the observation that mLST8 binds to the disordered regions of the N- and C-termini of the CCT subunits.

The intermolecular cross-links were also valuable in docking mLST8 within the cryo-EM density between the CCT rings. The circular shape of the mLST8 β-propeller allows it to fit the density in several orientations. However, we were able to identify a preferred orientation by minimizing cross-linking distances in the docking (Fig. 5a). Since the links involved the disordered termini of CCT subunits not resolved in our structural model, we used

distance constraints to the last ordered residue of the corresponding termini (CCT1 D528, CCT3 S17, CCT4 P30, and CCT6 V13) in the analysis. The docking positioned the mLST8 β-propeller with the K215 side facing the center of the folding cavity.

The intramolecular mLST8 cross-links further support the structural observation that mLST8 has achieved a near-native structure while bound to CCT. When the eight mLST8 intralinks were mapped onto the atomic structure of mLST8[28], all but two fell within the 32 Å distance constraint (Fig. 5b and Supplementary Table 3) when there are 10 of 28 possible cross-links in the mLST8 crystal structure that exceed the distance constraint. If mLST8 were less folded and highly flexible while bound to CCT, we would have expected more cross-links incompatible with the crystal structure. These observations suggest that at this late stage of folding by CCT, mLST8 adopts a limited ensemble of structures that closely resembles the native state.

**The nucleotide state of CCT in the mLST8–CCT complex**. As described above, the mLST8–CCT complex was purified directly from cells without adding exogenous nucleotide, so the CCT oligomer should only contain tightly bound nucleotide that has withstood the purification process. The structure of a yeast substrate-free CCT also purified in the absence of added nucleotide was recently reported[13]. This structure showed that the nucleotide-binding pocket was empty in five of the subunits, but was fully occupied in CCT6 and CCT8 and partially occupied in CCT3. The authors termed this state the nucleotide partially preloaded state (NPP). To determine the nucleotide-binding state of mLST8–CCT, we generated a difference map of the equatorial region of the mLST8–CCT complex and that of the AMP–PNP-bound state of yeast CCT[13], which contains nucleotide in all 16 nucleotide-binding sites (Fig. 6a). The difference map would, therefore, show which nucleotide-binding sites were occupied in the mLST8–CCT complex (no electron density difference) and those that were unoccupied (an electron density difference). Based on these differences, all the nucleotide-binding sites were empty except the two CCT8 subunits, CCT6 and partially in CCT6′ (Fig. 6a). This nucleotide site occupancy is similar to that of yeast except for CCT3, which was partially occupied in yeast NPP CCT[13] and empty in human mLST8–CCT. These findings show that subunits with residual nucleotide-binding and low-ATP utilization reside on the CCT6 side of the ring, roughly the same side that binds mLST8.

Further examination of the nucleotide-binding pocket of either of the CCT8 subunits revealed electron density that is clearly attributable to ADP (Fig. 6b). The ADP molecule has 93% of its

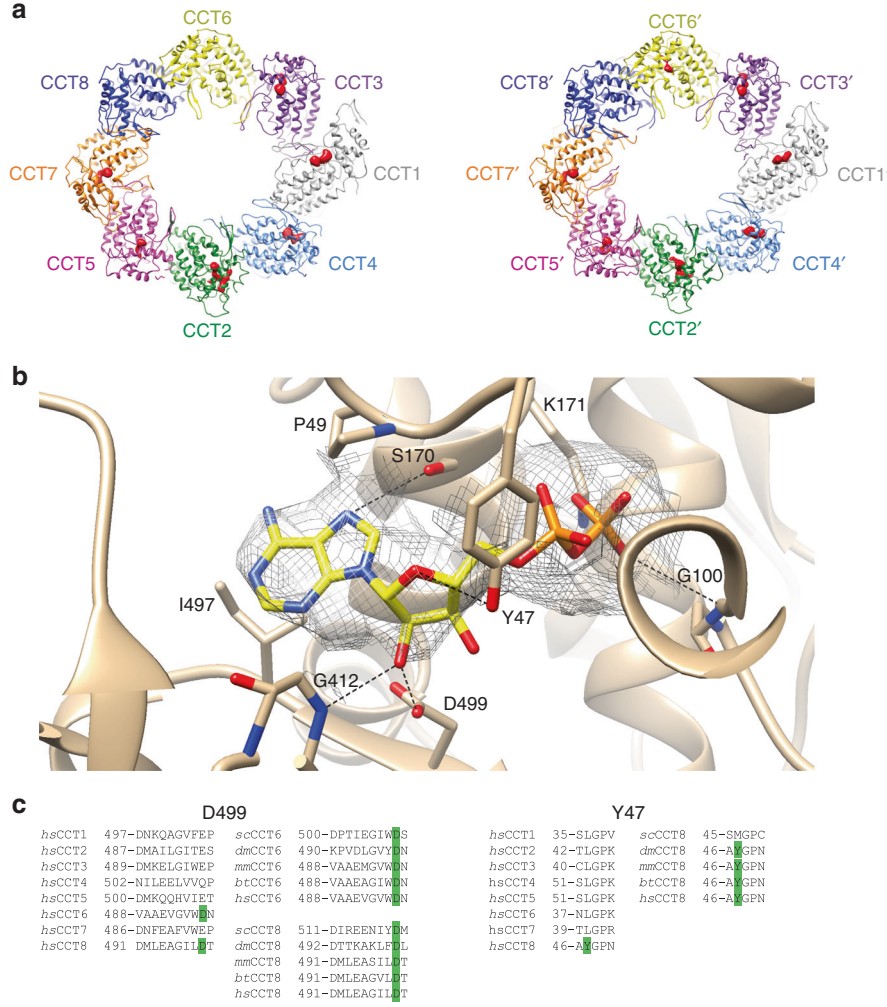

**Fig. 6** Nucleotide-binding state of the CCT–mLST8 complex. **a** Distribution of nucleotide density in the difference map between the yeast AMP–PNP CCT and the human mLST8–CCT reconstructions. Since all the nucleotide-binding sites of the AMP–PNP CCT reconstruction host a nucleotide, density differences (highlighted in red) indicate unoccupied nucleotide-binding pockets in mLST8–CCT complex, and lack of red density indicates the presence of nucleotide in CCT8, CCT8′, CCT6 and partially in CCT6′. **b** ADP binding site in CCT8. The protein ribbon diagram is shown in tan with residues in close contact with the ADP molecule highlighted as sticks. The ADP is also represented in sticks with carbon in yellow, oxygen in red, nitrogen in blue and phosphorous in orange. Hydrogen bonds are depicted as dashed lines. The map contour level is set at 2.5. The figure was produced with Chimera. **c** Alignments of the CCT amino-acid sequences adjacent to D499 and Y49 of CCT8 comparing the eight human CCT subunits and CCT6 and CCT8 from different species. Residues identical to D499 of CCT6 and CCT8 and Y49 of CCT8 are highlighted in green. *hs* Homo sapiens, *bt* Bos taurus, *mm* Mus musculus, *dm* Drosophila melanogaster, *sc* Saccharomyces cerevisiae

solvent accessible area (566.6 Å$^2$) buried by contacts with the interacting residues. There are several residues that are positioned to make important hydrogen bonds (Fig. 6b). In addition, K171 is positioned to form a salt bridge with the β phosphate. Hydrophobic interactions also contribute to the interaction with P49 and I497 flanking the adenine base on either side. All of these residues are conserved among the eight human CCT subunits. Less conserved interactions may explain why CCT6 and CCT8 release ADP more slowly than the other subunits. D499 is unique to CCT6 and CCT8 and sits near the hydroxyl groups of the ribose ring at close hydrogen bonding distance. In the other CCT subunits, this position is occupied by E or Q (Fig. 6c). The additional length of these side chains would cause steric clashes with the ribose ring, forcing a repositioning of ADP that could decrease its binding affinity. Furthermore, D499 is conserved in CCT6 and CCT8, supporting the idea that this residue is important in high affinity ADP binding. Y47 is another residue unique to CCT8 that is in position to hydrogen bond with the

ribose ring oxygen. All other human CCT subunits have a leucine at that position (Fig. 6c), which is unable to form the hydrogen bond, suggesting that Y47 also contributes to the higher affinity binding of ADP to CCT8.

**Comparison with yeast CCT**. Despite the similarities in nucleotide occupancy, a comparison of the mLST8–CCT structure with that of yeast NPP–CCT revealed a notable difference in the CCT2 apical domain. In both structures, the chaperonin assumes an open conformation with very similar structures in the equatorial and intermediate domains. However, in yeast NPP–CCT the intermediate and apical domains of CCT2 adopt a Z-shaped conformation in which its helical protrusion projects sharply outward away from the CCT folding cavity[13] (Supplementary Fig. 8). This conformation was not observed in the mLST8–CCT structure, which shows the CCT2 apical domain tilted slightly inward toward the center of the folding cavity like the other CCT subunits (Supplementary Fig. 8). This

conformational difference cannot be explained by the presence of substrate because the Z-shape is not observed in substrate-free human CCT either (Supplementary Fig. 7). Moreover, a recent 8 Å structure of bovine CCT shows a similar conformation in the CCT2 apical domain as mLST8–CCT[32]. Thus, the Z-shaped conformation appears to be unique to the yeast CCT2 apical domain.

Addition of AMP–PNP to yeast CCT changed the conformation of the CCT2 subunit to one very similar to mLST8–CCT (Supplementary Fig. 6a), but closer inspection revealed that several of the apical domains of mLST8–CCT were tilted more inward and downward toward the center of the folding cavity than with yeast AMP–PNP–CCT, partially closing the folding cavity. These differences were generally greater in those subunits interacting with mLST8 (Supplementary Fig. 6b), suggesting that this partially closed conformation may be caused by mLST8 binding. However, the conformational changes would have to be communicated allosterically from the mLST8 binding site between the rings because there are no direct interactions between the CCT apical domains and mLST8. Such long-range conformational changes are known to occur in CCT when ATP hydrolysis in the equatorial domains results in changes in the apical domains that close the folding cavity[21].

## Discussion

The findings reported here provide evidence that the cytosolic chaperonin CCT contributes to mTOR complex assembly and mTOR signaling by folding the β-propellers of mLST8 and Raptor, affording a possible explanation for the genetic links between CCT and yeast TOR observed previously[7–9]. The high-resolution structure of the mLST8–CCT complex provides insight into how mLST8 folding may occur. The mLST8 has achieved a near-native state while bound to CCT, suggesting that the complex represents a late folding intermediate that is ready to bind mTOR upon release from CCT. The position of mLST8 deep in the CCT structure between the rings is surprising because this region has not previously been implicated in substrate binding. Most studies of both type I and type II chaperonins have identified substrate binding sites in the apical domains far from the N- and C-termini in each ring[12,32,46–49]. These termini are all located at the bottom of the folding cavity and have been proposed to create a barrier that separates the two folding cavities[19,40]. However, the mLST8–CCT structure shows that the folding cavities are not separated and that CCT can bind substrates between the rings via interactions with the N- and C-termini. This observation is consistent with other biochemical and structural studies suggesting that the termini can participate in substrate interactions for both type I[50–52] and type II[27] chaperonins. These interactions appear to involve specific substrates and/or late stage intermediates in the folding process.

A closer look at the mLST8–CCT reconstruction reveals several thin masses that extend from the equatorial domains of CCT5, CCT7, CCT8, CCT6, and CCT3 in one ring and CCT5′, CCT7′, CCT8′, CCT6′, and CCT1′ in the other ring to suspend mLST8 between the rings (Fig. 4b). These masses likely correspond to the N- and C-termini of the subunits because they are known to extend into the space between the rings[31,40] and several of the termini cross-link to mLST8 (Fig. 5a and Supplementary Table 3). The disordered nature of these regions does not allow a more detailed description of the interactions, but a difference map between the mLST8–CCT reconstruction and the structure of native mLST8 docked into the mass between the rings shows extra mass, attributable to the termini, that surrounds the bound mLST8 at specific points

(Supplementary Fig. 9). These observations suggest that CCT contacts mLST8 almost exclusively through the termini of the subunits.

CCT is known to fold the β-propellers of other proteins, including G protein β subunits. A previous, low-resolution cryo-EM structure of the Gβ1–CCT complex, also purified directly from cells[18], shows that despite its close structural homology to mLST8, Gβ1 interacts with CCT in the apical domains where other CCT substrates have been shown to bind (Fig. 7a). A question that arises from these observations is how CCT interacts so differently with structurally homologous β-propellers like Gβ and mLST8. The answer may lie in the N-terminal α-helix of Gβ, not found in mLST8, which makes a coiled-coil interaction with the G protein γ subunit in the Gβγ dimer. This helix contacts the apical domain of CCT3 and holds Gβ high in the CCT folding cavity where it can interact with PhLP1, the CCT co-chaperone that releases Gβ from CCT to form the Gβγ dimer[18]. In contrast, PhLP1 does not assist in mLST8 folding or release (Supplementary Fig. 3), probably because PhLP1 cannot access mLST8 between the CCT rings from its binding site at the top of the CCT apical domains. Instead, ATP binding and hydrolysis contribute to release of mLST8 from its position between the rings (Fig. 1g), suggesting that the conformational changes in CCT upon ATP hydrolysis dislodge mLST8 to interact with mTOR. Interestingly, the mTOR binding site on mLST8 is exposed in the interior of the CCT folding cavity (Fig. 7b), suggesting that the mTOR kinase domain could interact with mLST8 while still bound to CCT.

In the mLST8–CCT structure, mLST8 is located on the CCT6 hemisphere of the ring (Fig. 4a), associating principally with CCT5, CCT7, CCT8, and CCT6. Gβ also associates with the CCT6 side, despite binding to the apical domains[18]. Interestingly, the CCT6 and CCT8 subunits on this same side of the ring retain their nucleotide throughout the tandem affinity purification . This slow release from the CCT6 hemisphere explains previous observations that showed poor ATP binding on the CCT6 side[43]. An asymmetric ATP-binding and sequential substrate folding mechanism for CCT was previously proposed in which bound substrates are first released from the CCT2 hemisphere, because of efficient ATP binding and hydrolysis on the CCT2 side, and are then retained on the CCT6 hemisphere because of low-ATP utilization[43,53]. The positions of mLST8 and Gβ in folded β-propeller structures on the CCT6 side is consistent with this sequential folding mechanism. It is possible that sequential release of β-propeller proteins may facilitate their complex folding trajectory, which requires that the N- and C-termini come together to form the last blade of the β-propeller.

These results reveal an important function for CCT in the folding of mLST8 and Raptor in preparation for their assembly into mTOR complexes. Based on our findings and other studies on protein folding by CCT, we propose a hypothetical scheme for the assembly of mTOR complexes (Fig. 7c). The β-propeller domains of nascent mLST8 and Raptor likely bind to CCT in a partially folded state either co-translationally or soon thereafter[12]. They are then folded by CCT into their β-propeller structures through cycles of ATP binding and hydrolysis and are released to interact with nascent mTOR and create the mTORC1 complex, while mTOR itself is folded by the Hsp90 TTT-R2TP co-chaperone complex[5,6]. In the case of mTORC2, mLST8 and mTOR are folded by the same mechanism, but it is currently not known how the other core components, Rictor and mSIN1, are folded before assembly. There are a number of questions yet to be answered in mTORC assembly, but this study establishes a key role for CCT in the process. These contributions of CCT to mTORC assembly may underlie the diseases caused by inactivating CCT mutations[54] or the increased CCT activity in cancer

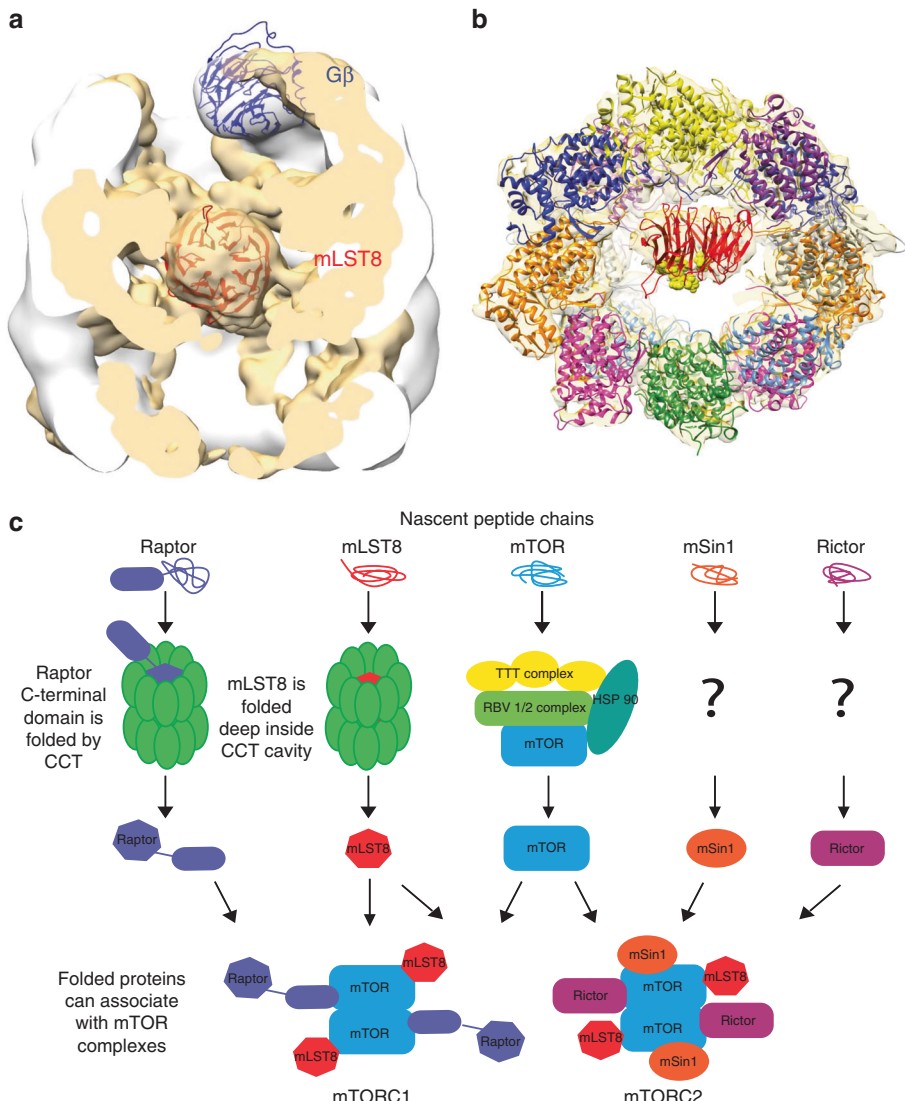

**Fig. 7** The role of CCT in the folding of β-propeller proteins. **a** Differences in the β-propeller location in the CCT cavity. The 3D reconstructions of Gβ–CCT (white)[18] and mLST8–CCT (tan) reveal the different location of the two β-propellers. **b** Position of the mTOR binding site on mLST8 in the mLST8–CCT complex. Residues on mLST8 that bind mTOR are shown as yellow spheres. **c** Hypothetical scheme of the chaperones involved in assembly of mTOR complexes, highlighting CCT-dependent folding of mLST8 and Raptor

cells[55], given the essential functions of mTOR in regulating cell metabolism, growth, and survival.

## Methods

**Cell culture**. Human embryonic kidney (HEK)-293T cells (ATCC) were grown in 10% fetal bovine serum (FBS) in DMEM/F12 media, and HepG2 liver hepatocellular carcinoma cells (ATCC) were grown in DMEM media. Cells were passaged to maintain confluency between 10–90% and passage number was kept under 15.

**CRISPR knockdown**. Knockdowns were done using the PX459 V2.0 vector (Addgene) containing sgRNAs targeting CCTe, Raptor or PhLP1. The same vector with a nontargeting sgRNA segment was used as a negative control. The sgRNA sequences are provided in Supplementary Table 4. These vectors were transfected into HEK-293T cells at 25–40% confluency with Lipofectamine 3000 (Thermo Fisher Scientific) and then treated with 1 μg/mL puromycin (Invivogen). Forty-eight hour after transfection of PX459V2.0, cells were transfected with vectors containing mTORC or Gβγ components. Cells were harvested for immunoprecipitation and immunoblotting 96 h after addition of the sgRNA vector.

**Immunoprecipitation**. HEK-293T cells were cultured and transfected in six-well plates. Cells were washed in phosphate-buffered saline (PBS) and lysed in 200 μL of PBS supplemented with either 1% IGEPAL (for mTORC1 and Gβγ) or 0.3–0.5% CHAPS (for CCT and mTORC2), 0.5 mM PMSF and Halt Protease Inhibitor

Cocktail (Sigma P8340). Protein concentrations were determined using the DC protein assay (BioRad 5000116) and equal protein amounts (~400 μg) were immunoprecipitated by addition of epitope tag antibodies according to Supplementary Table 5, followed by 30 μL of protein A/G agarose beads (Santa Cruz). Immunoprecipitants were washed three times in lysis buffer, then resuspended in sodium dodecyl sulphate polyacrylamide gel electrophoresis (SDS-PAGE) loading buffer. Proteins were separated by SDS-PAGE and transferred to nitrocellulose (Biorad Transblot). The nitrocellulose was probed with the indicated primary antibodies and IRDye secondary antibodies (Li-COR) at the dilutions indicated in Supplementary Table 5. Blots were imaged using a LI-COR Odyssey infrared scanner, and proteins were quantified with the LI-COR software.

The effects of ATP on the co-immunoprecipitation of mLST8 and Raptor with CCT was performed in a cell line expressing a Flag-tagged CCT3 subunit. To insert the Flag tag, we transfected HEK293T cells with Cas9 and sgRNA targeting CCT3 (Addgene px458 vector) along with a double stranded DNA fragment for a donor template containing a Flag tag to be inserted in an external loop between P374 and K375 (synthesized G block by Integrated DNA Technologies). Cells were sorted into 96-well plates and monoclonal lines were screened via immunoblotting for a Flag-tagged CCT3. The cell line was verified by PCR and sequencing. These cells were transfected with either mLST8 or Raptor. We then performed an immunoprecipitation as described above using Flag antibody. After two washes in lysis buffer containing 0.5% CHAPS, samples were incubated at 4 °C in this buffer for an additional 3 h during which 5 mM ATP was added after 0, 1, or 2 h so that samples were incubated in ATP for a total of 1, 2, or 3 h. The zero sample was incubated without ATP for 3 h.

Following the incubation, samples were washed two more times and then immunoblotted as previously described.

**Insulin signaling**. HepG2 cells were chosen for these experiments because of their robust response to insulin. Cells were treated with 40 nM of siRNA targeting CCTα/ε, mLST8 or Raptor or a non-targeting control using Lipofectamine 3000. Seventy-two hour after knockdown, cells were serum starved for 18 h. Subsequently, the cells were treated with 100 nM insulin for 30 min and the lysates were harvested in 1% IGEPAL in HEPES buffered saline supplemented with 0.5 mM PMSF, Halt Protease Inhibitor Cocktail and phosphatase inhibitor (Thermo Fisher). The effects on mTORC1 and mTORC2 signaling were analyzed by immunoblotting the cell lysates for phosphorylation at IRS1 S636/639 and AKT S473, respectively. Total CCT, mLST8, Raptor, AKT, and IRS1 were also immunoblotted to assess the respective knockdown and expression level. Antibody sources and dilutions used are indicated in Supplementary Table 5.

**Quantitative polymerase chain reaction**. HEK-293T cells were depleted of CCT using CRISPR as described above and were harvested for RNA isolation (Zymo RNA isolation kit) 96 h later. Qiagen one-step RT PCR kit was used for the reverse transcription step. The qPCR was done using IDT PrimeTime assays with predesigned primers for mTOR, Raptor, mLST8, and HPRT as a control (Supplementary Table 4). Real time PCR was performed using the QuantStudio 5 Real Time PCR system and data were analyzed using the QuantStudio Design and Analysis software.

**Statistical analysis**. The statistical analysis was performed using the BootstRatio, a web-based statistical analysis program which calculates probability that the relative expression, RE ≠1[56]. BootstRatio allows calculation of statistical significance when data are normalized to a control sample. The application can be found at http://regstattools.net/br.

**Isolation of mLST8–CCT**. HEK-293T cells were cultured as described above in T-175 flasks. At 80% confluency, each flask was transfected with 45 μg N-terminal HPC4-Twin Strep-Flag-mLST8 in pcS2+ vector and with 45 μg His6-myc-PhLP1 in pcDNA3.1B+ using 200 μg of polyethylenimine (PEI). The media was replenished with DMEM/F12 supplemented with 10% FBS after 2–4 h, and the cells were incubated for 48 additional hours before harvesting. Cells from each T175 were lysed in 2 mL of extraction buffer 1% IGEPAL in PBS with 0.5 mM PMSF and Halt Protease Inhibitor Cocktail. The lysate was cleared by centrifugation at 30,000×g for 20 min. The lysate was then filtered through a 0.45 μm and then a 0.2 μm filter.

The mLST8–CCT complex was purified at 4 °C by tandem affinity purification. The filtered lysate was loaded for 1 h onto a HisTrap HP 5 mL column (GE17-5248-01) equilibrated with 20 mM HEPES, 20 mM NaCl, 25 mM imidazole, 0.05% CHAPS, 1 mM TCEP, pH 7.5. The column was washed with 5 column volumes of equilibration buffer. A linear gradient from 25 to 500 mM imidazole was then applied over 8 column volumes. Elution fractions were then analyzed by SDS-PAGE and Coomassie staining. Fractions containing mLST8 and CCT were combined and loaded for 1 h onto 5 mL of Strep-Tactin resin (Iba 2-1201-010) equilibrated with 20 mM HEPES pH 7.5, 20 mM NaCl. The column was washed twice with one column volume of 20 mM HEPES pH 7.5, 150 mM NaCl, 0.05% CHAPS, and then twice with one column volume of 20 mM HEPES pH 7.5, 20 mM NaCl, 0.05% CHAPS. The mLST8–CCT complex was eluted with 3 column volumes of 20 mM HEPES pH 8.0, 20 mM NaCl, 0.05% CHAPS, 2.5 mM D-desthiobiotin and concentrated to 1 μg/μL using a 30 kDa cutoff filter (Amicon UFC803024) and analyzed by SDS-PAGE and Coomassie staining and immunoblotting. The sample was then flash frozen in liquid nitrogen.

**Isolation of substrate-free CCT**. HEK-293T cells were cultured in T-175 flasks as described above. At 80% confluency, each T175 flask was transfected with 90 μg of His6-myc-PhLP1 in pcDNA3.1B+ using 200 μg of PEI. The cells were then lysed and loaded onto a HisTrap column as described for mLST8–CCT. The combined HisTrap elution fractions were then loaded for half an hour onto a HiTrap Heparin HP 5 mL column (GE17-0406-01) equilibrated with 20 mM Tris-HCl pH 7.5, 150 mM NaCl, 2.5 mM MgCl2, 1 mM TCEP. The column was washed with two column volumes of equilibration buffer. A linear gradient from 150 mM to 1 M NaCl was then applied over 8 column volumes. Elution fractions were then analyzed by SDS-PAGE and Coomassie staining, and fractions containing CCT were combined and concentrated in a 30 kDa cutoff filter (Amicon UFC803024) to less than 300 μL. The sample was then injected onto a Superose 6 10/300 GL size exclusion column (SEC) equilibrated in 20 mM HEPES pH 7.5, 150 mM NaCl, and 1 mM TCEP mobile phase. One column volume of mobile phase was then run over the column and fractions were analyzed by SDS-PAGE and Coomassie staining. The sample was concentrated to 1 μg/μL using a 30 kDa cutoff filter and flash frozen in liquid nitrogen.

**Cross-linking**. Approximately, 200 μg of mLST8–CCT complex was cross-linked in 25 mM HEPES pH 8.0, 100 mM KCl, and 325 μM of a 50% mixture of H12/D12 DSS (Creative Molecules) at 37 °C for half an hour. The reaction was quenched by

adding 50 mM ammonium bicarbonate to the cross-linked sample and incubating at 37 °C for 15 min. The sample was dried using a vacuum concentrator, denatured in 100 mM Tris-HCl pH 8.5 and 8 M urea, reduced with 5 mM TCEP at 37 °C for 30 min, and alkylated with 10 mM iodoacetamide at room temperature in the dark for 30 min. The sample was diluted with 150 mM ammonium bicarbonate to bring the urea concentration to 4 M, and proteins were digested with 4 μg of lysyl endopeptidase (Wako 125-05061) (1:50 enzyme: substrate ratio) at 37 °C for 2 h. Subsequently, the urea was diluted to 1 M, and proteins were further digested with trypsin (Promega V5111) at a 1:50 ratio at 37 °C overnight. Peptide fragments were purified on a C18 column (Waters WAT054955), dried, and reconstituted in 35 μL SEC mobile phase (70:30:0.1 water:acetonitrile:TFA). Cross-linked peptide fragments were enriched by SEC using a Superdex Peptide PC 3.2/30 column at a flow rate of 50 μL/min. The fractions with the highest peptide concentration were dried and resuspended in 2% formic acid.

**Mass spectrometry**. The enriched cross-linked peptide samples were separated using a Thermo Fisher Scientific EASY-nLC 1000 Liquid Chromatograph system with a 15 cm Picofrit column (New Objective) packed with Reprosil-Pur C18-AQ of 3 μm particle size, 120 Å pore size and gradient of 5–95% acetonitrile in 5% DMSO and 0.1% formic acid over 185 min and at a flow rate of 350 μL/min. The column was coupled via electrospray to an Orbitrap Velos Pro mass spectrometer. The resolution of MS1 was 30,000 over a scan range of 380–2000 m/z. Peptides with a charge state +3 and greater were selected for HCD fragmentation at a normalized collision energy of 35% with 3 steps of 10% (stepped NEC) and a resolution of 7500. Dynamic exclusion was enabled with a 10 ppm mass window and a 1-min time frame. Samples were run in duplicate.

**XL-MS analysis**. The XL-MS spectra were analyzed using the pLink 2 software suite[45]. First, the peptide sequence database was created from the amino-acid sequence of human mLST8 and the eight human CCT subunits (UniProt ID: Q9BVC4, P17987, P78371, GenBank: CAA52808.1, P50991, P48643, P40227, Q99832, P50990, respectively). A database of 293 common contaminant proteins were also added by pLink. The program was then run using the preset DSS conventional cross-linking (HCD) conditions and a custom heavy DSS linker profile (LinkerComposition: C(5)H(−2)2H(6)O(2), MonoComposition: C(5)H(6)O(3), LinkerMass: 102.064, MonoMass: 120.075), with trypsin set as the protease and up to 3 missed cleavages allowed. Peptides were selected with a mass between 600 and 6000 Da and a length between 6 and 60 amino acids. The precursor and fragment tolerances were ±20 ppm. The peptides were searched using carbamidomethyl (C) fixed modifications and phospho Y, T, S, and oxidated M variable modifications. The results were filtered with a filter tolerance of ±10 ppm and less than 5% FDR.

**Cryo-EM grid preparation and data acquisition**. Cryo-EM grids were prepared with a Vitrobot Mark IV (FEI) at 22 °C and 95% humidity. Aliquots of 4 μl of human CCT–mLST8 complex or apo–CCT were applied to a glow discharged holey carbon grid (Quantifoil R2/2, 300 mesh). Grids were previously treated with polylysine to increase the number of side views of the chaperonin, as previously described[13]. The mLST8–CCT data acquisition were performed at ESRF Grenoble with a FEI Titan Krios electron microscope (Krios 1) operating at 300 kV, equipped with a Gatan K2 Summit direct electron detector mounted on a Gatan Bioquatum LS/967 energy filter. Data collection was carried out with a nominal magnification of ×105,000 (yielding a pixel size of 1.36 Å/pixel), at a defocus range of −1.5 to −3.0 μm. A total of 5576 movies were recorded and fractionated to 40 frames with a total exposure of 7 s. The dose rate was 5.2 e−/pixel/s for a total dose of 36 e−/Å2 on the specimen.

Substrate-free CCT grids were prepared as described above and images were collected on a FEI Titan Krios electron microscope operating at 300 kV at Diamond Light Source electron Bio-Imaging Centre (eBIC) (Krios 1), at a nominal magnification of ×130,000(corresponding to a pixel size of 1.06 Å/pixel). A total of 2293 movies (40 frames/movie) were recorded on a Gatan Quantum K2 Summit direct electron detector operated in counting mode, with a defocus range of −1.5 to −3.0 μm. Each movie was exposed for 8 s, with an exposure rate of 5.3 e−/pixel/s, leading to a total accumulated dose of 42 e−/Å2 on the specimen.

**Image processing**. The 5576 movies of mLST8–CCT complex were aligned using MotionCorr2[57] program as part of the Scipion processing workflow[58]. The MotionCorr2 output was subjected to CTF determination using CTFFIND4[59]. Totally, 1,769,600 particles were automatically picked with Xmipp[60] and extracted with a downsampling factor of 3 (4.08 Å/pixel, 68 pixel box size). All the image processing steps were carried out without any symmetry imposition. A first 2D classification using Relion 2.0[61] (unless otherwise stated Relion 2.0 was used in all subsequent steps) was performed to exclude bad particles and ice contamination. Some of the best 2D classes were used as a template to generate an initial model using both EMAN[62] and RANSAC[63]. In both cases a cylinder with the general dimensions similar to the CCT structure was obtained, which was subsequently used for the iteration process. One of the models was low-pass filtered to 60 Å and used for a 3D classification of 1,197,358 particles contained in the best 2D classes. The 3D classes that showed well-defined CCT features and a mass inside the cavity

(860,453 particles) were subjected to refinement using 3D auto-refine, which generated a 9 Å map. The particles used in this refinement were re-extracted from the 1.36 Å/pixel micrograph to continue the processing with the original data. A new 3D classification was performed in which a mask was applied around the mass attributed to the substrate in order to favor the classification to the substrate contribution and prevent the CCT predominance. Those particles with a better-defined mLST8 mass (452,000) were finally subjected to auto-refine and a map was obtained with a final resolution of 4.35 Å. Subsequently, a postprocessing was performed masking the previous map and enhancing the high frequencies and the resolution improved to 4.0 Å, as estimated using the gold standard FSC criterion at 0.143[64]. Local resolution in the 3D structure of the mLST8–CCT complex was estimated using MonoRes[65] from Xmipp package and ResMap[66]. The statistical information is listed in Supplementary Table 1.

The 3D reconstruction of substrate-free CCT was carried out following a similar procedure. A total of 2293 movies were aligned with MotionCor2 and CTF corrected. A total of 504,060 particles were automatically selected and extracted with a downsampling factor of 3 (3.18 Å/pixel, 80 pixel box size). Particles were 2D classified using Relion 2.0 and the best classes (139,819 particles) were subjected to 3D classification. Classes with the best structural features of CCT were further refined, the particles were re-extracted from the original micrographs (1.06 Å/pixel) and after 3D refinement, a final map at 7.5 Å was obtained (gold standard FSC).

**Model building.** Models for each human CCT chain were generated using SWISS-MODEL homology-modeling server[67] using the yeast CCT structure (PDB 5GW5) as a reference. The resulting model was docked into the cryo-EM density using Chimera[68] and further subjected to flexible fitting of the individual subunits with iMODFIT[37]. Manual adjustment and real-space refinement were carried out in COOT[69] to increase the quality of the fitting. The resulting model was refined by several rounds using PHENIX[38] and CCP-EM[70] software suites. The restraints used in phenix real-space refinement were both the standard (bond, angle, planarity, chirality, dihedral and nonbonded repulsion), with some additional restraints (Ramachandran plot, C-beta deviations, rotamer, and secondary structure). A local grid search-based fit was included in the refinement strategy to resolve side-chain outliers (rotamers or poor map fitting). Validation of the final model was done using the phenix.validation_cryoem module in PHENIX. The final refinement statistics are provided in Supplementary Table 2.

**Nucleotide distribution analysis.** In order to detect and identify nucleotides in the CCT subunits, the equatorial domain of yeast CCT ATP-BeF₂ (PDB 4D8R) was docked into the human CCT–mLST8 model and the yeast CCT AMP–PNP model (PDB 5GW5) in order to segment the equatorial domains. A difference map between them was generated using the vop subtract operation implemented in Chimera. A detailed inspection of the nucleotide-binding pocket of all the CCT subunits of mLST8–CCT showed a clear difference between nucleotide-free and nucleotide-bound subunits and allowed the modeling of an ADP molecule in CCT8. The NCBI Blastp web-based suite was used to perform an alignment of the eight human CCT subunits (Uniprot ID: P17987, P78371, P49368, P50991, P48643, P40227, Q99832, and P50990) as well as yeast, drosophila, murine, and bovine CCT6 (Uniprot ID: P39079, Q9VXQ5, P80317, and Q3MHL7) and CCT8 (Uniprot ID: P47079, Q7K3J0, P42932, and Q3ZCI9) to identify conserved and unique residues in the ADP binding site.

**Reporting summary.** Further information on research design is available in the Nature Research Reporting Summary linked to this article.

## Data availability

Cryo-EM data have been deposited in the Electron Microscopy Data Bank under accession codes EMD-4489 and EMD-4503 for the human mLST8–CCT complex and human apo–CCT, respectively. The associated atomic model for mLST8–CCT has been deposited in the Protein Data Bank under accession code PDB 6QB8. The raw mass spectrometry data are available on the MassIVE repository under the ProteomeXchange accession code PXD013975 and the MassIVE accession number MSV000083839. The Source Data file contains the raw data for all graphs and the uncropped versions of immunoblots presented in the figures as well as the pLink cross-link identification output. Other data are available from the corresponding authors upon reasonable request.

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

## Acknowledgements

This research was supported by the grant BFU2016-75984 (AEI/FEDER, EU) and the Madrid Regional Government (grant S2013/MIT2807) to J.M.V. as well as the US National Institutes of Health grant EY012287 to BMW and fellowships from the Brigham Young University Simmons Center for Cancer Research to W.G.L., N.C.T., T.A. and M. D. We are grateful to the Diamond Light Source for access to its cryo-EM facility thanks to the proposal EM15997 and help from Dr. Kyle Dent. We also acknowledge the European Synchrotron Radiation Facility for microscope time on CM01 and we would like to thank Drs. Eaazhisai Kandiah and Gregory Effantin for their assistance.

## Author contributions

N.C.T, M.D., M.J.M. and R.L.P. performed the biochemical experiments; W.G.L. and T. A. purified the mLST8–CCT complex; W.G.L. and M.T.B-C. purified the apo–CCT complex, J.C. and M.T.B-C. acquired the cryo-EM images; J.C., M.T.B-C. and C.S. did the image processing; T.A, W.G.L, A.M., and S.F. did the cross-linking/mass spectrometry; J. C., W.G.L., N.C.T., M.T.B-C., T.A., M.D., C.S., J.M.V. and B.M.W analyzed the data; B. M.W., J.M.V., J.C., W.G.L. and N.C.T. designed the experiments and wrote the paper.

## Additional information

**Competing interests:** The authors declare no competing interests.

