## [Peer Review File · Nature Communications]

Reviewers' comments:

Reviewer #1 (Remarks to the Author):

This paper describes the first structure of human CCT and the highest resolution structure, to date, of a CCT-substrate complex. More importantly, it is the first structure in which the substrate is positioned between the 2 rings. The paper is well written and potentially very important for understanding CCT's functions. A revision should address the points below.

Comments

1. The discussion of the crosslinking data appears to assume the absence of an ensemble of conformations. However, it is possible that each of the constraints is satisfied only by some of the conformations. In such a case, it is possible that the substrate is unfolded. The crosslinking data does suggest the absence of nonnative interactions.
2. More importantly, the observation of the location of the substrates raises many questions that are, perhaps, beyond the scope of this work but should be acknowledged in the Discussion. These questions include the mechanism by which the protein reaches this location (is it threaded and what is the driving force?), the mechanism of release (does it involve ring separation?) and the reason(s) that the mLST8 substrate behaves differently from other substrates (functional and structural reasons).

Minor comments:

1. Refs. 12 and 24 are inappropriate where cited.
2. Figures 2, S2 and elsewhere – why are standard errors shown and not standard deviations?
3. p. 21, ref. 54 – I believe that the ATP-dependent sequential folding mechanism was proposed much before in *Nat Struct Mol Biol.* 2005 Mar; 12(3):233-7.

Reviewer #2 (Remarks to the Author):

This is an important advance in understanding the physical interactions between the CCT chaperonin and some WD40 protein components of the TOR signalling pathway. Another protein complex, the anaphase promoting complex (APC) also has regulators which are WD40 repeat proteins, CDH1 and CDC20, which also interact with CCT and almost no advances in understanding of any sort have been made since the interaction was discovered in yeast in 2003 (ref 27 *ibid*). There is some understanding of the PP2A WD40 - CCT interaction but at nowhere near the resolution of this data under review.

Generally speaking it would be fascinating to understand the numbers and the various modes of interaction between CCT and the huge WD40 protein family which has around 500 members in humans. This manuscript shows that LST8 interacts with CCT in an unexpected way; lying between the ring interfaces in a specific orientation.

1. Is the LST8 WD40 propellor closed completely or waiting to be sealed? We know that several yeast WD40 propellors are CCT-independent for folding (Willison, K.R (2018) The substrate specificity of eukaryotic cytosolic chaperonin CCT, *Philosophical Transactions of the Royal Society B-biological Sciences*, Vol: 373, ISSN: 0962-8436) and also that CDH1 cannot be released from CCT when mutations in the 'velcro' strands are present (Willison, K.R (2018)). What happens when LST8 is expressed in *E.coli* or by in vitro translation? If LST8 is expressed in insect cell expression systems and presented to CCT does it bind? I would like to see more focus on this issue. The nascent chain squiggles in Figure 6B are highly misleading in my opinion. The WD40 propellors must be significantly folded in order to interact with CCT specifically and with high affinity.
2. The introduction and discussion of the TOR-CCT interaction is shallow. There is a major omission in not discussing the work of Mike Hall's lab (PNAS 1996) which showed genetic interaction between CCT6 and the TOR pathway in yeast; especially since this new work shows that CCT6 interacts with LST8.
3. The negative result for PhLP1 interaction with the CCT-LST8 system is good to know but opens

up other questions. In particular what else is bound to the CCT-LST8 complex purified from the HEK293T cells (Supp fig S7)? The affinity purified material looks clean for CCT subunits; what is the occupancy by LST8? A good analysis would be to subject the CCT-LST8 complex to MALDI-MS analysis to ask what other proteins are bound in addition to LST8? The data may already exist from the X-link mass spec experiment.

4. A schematic summary of all the transfection data would be useful. Or better labelled gels with sketches above. It is hard work going through the individual gels (which I have done). This data is important though in ruling out what's in and what's out with respect to the CCT-TOR component interactions.

Reviewer #3 (Remarks to the Author):

General comments:

The paper is well written and the results provided should be of general interest to the field. The major claim of the paper is that CCT has an essential role in mTOR complex assembly. The Authors have used a number of experimental techniques to demonstrate that this is indeed the case. They have also isolated a late folding intermediate of mLST8 bound to CCT. Surprisingly the binding site for mLST8 was found to be deep within the folding chamber interacting with the N/C-termini of the equatorial domains rather than the apical domains. Cross-linking was used to confirm this binding site. This is an interesting result for the Group 2 chaperonins but I think some more discussion is need on what this means in relation to the group 2 and group 1 chaperonins. I think the paper is worth publishing and have made some specific comments below.

Specific comments:

Abstract:

Sentence 6: Please can the resolution claim be adjusted from 3.97 Angstroms to 4 Angstroms. The FSC resolution estimation is not accurate enough to go to the second decimal place.

Final sentence: "which is radically different from the rest of the chaperonins" This reviewer was under the impression that most of the eukaryotic group 2 chaperonins have sequential rather concerted ATP binding/hydrolysis mechanisms.

Results:

Page 7, Paragraph 2: the half life of the interaction seems very slow for both mLST8 and Raptor is this similar to other known CCT substrates.

Page 8: It would be interesting to know what happened to the unfolded mLST8 and Raptor in the CCT knockdown cells?

Page 11 Paragraph 2: What criteria did the authors use to select the 3D class that contained 452,000 particles from 1,197,358. Did the Authors look at the remaining classes?

Page 11, Paragraph 2: form the local resolution estimation it looks like the intermediate and apical domains are not as well resolved as the equatorial domains. Did the Authors try any further classification to see if a more isotropic reconstructions can be generated from a subset of the 452,000 particles?

Page 12, Final Paragraph: The mLST8 seems to fit well, however I think it would help the reader if the destiny was extracted for mLST8 and shown with the coordinates as it is difficult to truly asses the quality of fit when shown with the rest of the CCT structure. This could be a supplementary figure.

Page 13, First Paragraph: Was poly lysine used for the control CCT grids? Or did the authors not have a preferred orientation in the absence of substrate.

Page 13, The resolution of the control is much lower than that of the substrate bound complex?

There are 100,000 less particle but i am not sure that explains the difference. It would be good for

the authors to show the angular coverage they get from the CCT only structure and offer a potential reason for this.

Discussion:

Page 19, Paragraph 1: If the main binding site for the mLST8 is located at the N/C termini what is the role of the apical domains in binding/folding of mLST8? In an ideal world the authors would confirm the C/N termini as the primary binding sites by deletion studies. This has been done for GroEL, and has been shown to have an effect on folding efficiency of some substrates. I understand that making the mutants is a large undertaking and probably out of the scope of this study. However, I think some more discussion is needed and some comparisons to the group 1 chaperonins may help (Weaver and Rye, 2014).

Methods:

Page 32, final paragraph: for the Data Acquisition on mLST8-CCT the Authors say they collected 2753 movies but in the next page for the image processing they then say that they motion corrected 5576 movies. Please can this be clarified.

Page 33, Second paragraph: "The MotionCorr2 output was subjected to CTF correction using CTFFIND4" change to "The MotionCorr2 output was subjected to CTF determination using CTFFIND4"

Page 33, Second paragraph: Which initial model was used and why was it selected. I think some more detail may be useful.

Page 34, first paragraph: the local resolution estimation is not in Table S1. Could the authors provide the estimated local resolution (mean or median value) and the histogram for the resolution bins.

Page 34, 35, model building: What restraints did the authors use in phenix during real space refinement? As at this resolution refining non-bulky hydrophobic side chains could be difficult especially in the apical and intermediate domain regions.

Reviewer: Daniel Clare

Reviewer #4 (Remarks to the Author):

Cuéllar et al. provide exciting data on the folding of subunits of the mTOR complexes by the chaperonin CCT. Specifically, they obtained a high-resolution (~4 Å) cryoEM structure of the mLST8 subunit trapped inside the CCT folding chamber. In contrast to previous structures of CCT substrates, mLST8 is observed to be positioned deep inside the folding chamber, between the two rings of CCT.

The authors first show, using co-IP, that both mLST8 and a second mTOR complex subunit, Raptor, show strong interactions with CCT, lending support to the hypothesis that the beta-propellers of the two proteins are folded by CCT. This conclusion is further supported by cell viability assays, highlighting the role of CCT for mTOR complex assembly.

The cryoEM structure of mLST8 bound to CCT allowed a precise positioning of the substrate in the folding chamber: between the two rings and asymmetrically located towards subunits 3,6 and 8 of CCT, i.e. in the "CCT6 hemisphere" that exhibits weaker ATP binding and hydrolysis. Additionally, the authors were able to observe ATP occupancies in the structure of the mLST8-CCT complex. In addition to EM, the authors also performed crosslinking/mass spectrometry (XL-MS) experiments that provided some crosslinks on mLST8 itself and between mLST8 and CCT subunits, although overall I do not consider the XL data very helpful in support of the EM data (see specific comments below).

Together, Valpuesta et al. present very interesting results on the folding of mTORC subunits by CCT, providing new insights into the chaperonin's role in TOR pathways. The data will certainly help to better understand of the different ways that CCT can fold its substrates, as in this case a different positioning compared to other beta-propeller containing proteins is observed and a

connection to ATP binding and hydrolysis can be made.

Specific comments related to XL-MS data:

The evidence from XL-MS as presented on pages 14-15 and in Figure 4 is weak. Obtaining a sufficient number of crosslinks on a substrate trapped in the folding chamber is certainly challenging as a result of the timescale of folding and the general preference of forming crosslinks on exposed residues, most probably on the outer surface of CCT. The crosslinking results are certainly consistent with the positioning of mLST8 as derived from cryo-EM, but would fit to many other alternative locations of the substrate equally well because they only connect only a single residue of mLST8 (Lys 215) to several TRiC subunits. So, even if the substrate would be flipped to the other side of the folding chamber and predominantly interact with subunits of the other hemisphere, then the distance restraints would likely be fulfilled, at least judging from Fig. 4c. In this context, what does "high-quality crosslink" mean in the figure legend (page 39)?

It is also somewhat surprising that most of the crosslinks observed on mLST8 itself are formed across the propeller (Fig. 4d), bridging relatively large distances, compared to the linking of more adjacent Lys residues (that are involved in crosslinks across the molecule). Can this be explained by the specific orientation of the lysines on the beta-sheets?

In case the authors decide to keep the XL data in the manuscript, the relevance for making conclusions about the positioning/orientation of mLST8 need to be toned down a bit and more information than the selected few crosslinks listed in Table S3 needs to be provided. All crosslinks, also those on CCT, need to be listed together with essential data such as the exact peptide sequences that were identified, confidence scores, mass errors etc.

Reviewed by Alexander Leitner, ETH Zurich, Switzerland

Response to Critique

The reviewers' comments are in italics and our responses are in blue.

Reviewer #1 (Remarks to the Author):

This paper describes the first structure of human CCT and the highest resolution structure, to date, of a CCT-substrate complex. More importantly, it is the first structure in which the substrate is positioned between the 2 rings. The paper is well written and potentially very important for understanding CCT's functions. A revision should address the points below.

Comments

1. The discussion of the crosslinking data appears to assume the absence of an ensemble of conformations. However, it is possible that each of the constraints is satisfied only by some of the conformations. In such a case, it is possible that the substrate is unfolded. The crosslinking data does suggest the absence of nonnative interactions.

Although in principle the reviewer is correct, important observations indicate that the number of ensembles is not large and that they resemble closely the native conformation. First, the mass attributable to mLST8 between the CCT rings fits the crystal structure of mLST8 well (Figure 4). Second, the crosslinks are consistent with the crystal structure, with only two that we identified exceeding the crosslink distance limit out of 10 possible crosslinks that would have exceeded the limit (Figure 5b). We have modified the text in the last paragraph of the crosslinking section (pp. 14-15) to reflect these ideas.

2. More importantly, the observation of the location of the substrates raises many questions that are, perhaps, beyond the scope of this work but should be acknowledged in the Discussion. These questions include the mechanism by which the protein reaches this location (is it threaded and what is the driving force?), the mechanism of release (does it involve ring separation?) and the reason(s) that the mLST8 substrate behaves differently from other substrates (functional and structural reasons).

The reviewer raises some significant questions that are beyond the scope of this work, and we hesitate to speculate too much about them in the Discussion. However, we did discuss a way that mLST8 might reach its location on CCT through a sequential folding mechanism (p. 20). We also mentioned the role that ATP appears to play in mLST8 release and have added an observation that the mTOR binding site on mLST8 is exposed while bound to CCT, suggesting that mTOR may bind mLST8 while it is still bound to CCT (p. 19). Finally, we discuss in detail the differences between G β ₁ and mLST8 binding to CCT, specifically suggesting that the N-terminal helix of G β ₁, not found in mLST8, is responsible for the G β ₁ interaction with the apical domains of CCT (p. 19).

Minor comments:

1. Refs. 12 and 24 are inappropriate where cited.

We removed reference 12 (Willardson and Tracy) and moved reference 24 (Skjaerven et al.) to replace it. We also added Lopez et al. to reference that same sentence. We now believe these statements are appropriately cited.

2. Figures 2, S2 and elsewhere – why are standard errors shown and not standard deviations?

We believe that the standard error is appropriate when comparing average values of sample versus control to assess whether they are statistically different, which was our purpose here.

3. p. 21, ref. 54 – I believe that the ATP-dependent sequential folding mechanism was proposed much before in Nat Struct Mol Biol. 2005 Mar;12(3):233-7.

We added this reference to the sentence on p. 20 and apologize for any oversight.

Reviewer #2 (Remarks to the Author):

*This is an important advance in understanding the physical interactions between the CCT chaperonin and some WD40 protein components of the TOR signalling pathway. Another protein complex, the anaphase promoting complex (APC) also has regulators which are WD40 repeat proteins, CDH1 and CDC20, which also interact with CCT and almost no advances in understanding of any sort have been made since the interaction was discovered in yeast in 2003 (ref 27 *ibid*). There is some understanding of the PP2A WD40 - CCT interaction but at nowhere near the resolution of this data under review. Generally speaking it would be fascinating to understand the numbers and the various modes of interaction between CCT and the huge WD40 protein family which has around 500 members in humans. This manuscript shows that LST8 interacts with CCT in an unexpected way; lying between the ring interfaces in a specific orientation.*

1. Is the LST8 WD40 propellor closed completely or waiting to be sealed? We know that several yeast WD40 propellers are CCT-independent for folding (Willison, K.R (2018) The substrate specificity of eukaryotic cytosolic chaperonin CCT, Philosophical Transactions of the Royal Society B-biological Sciences, Vol:373, ISSN:0962-8436) and also that CDH1 cannot be released from CCT when mutations in the 'velcro' strands are present (Willison, K.R (2018).

The excellent fit of the atomic structure of mLST8 into the cryo-EM density (new Fig. 4c) and the consistency of the crosslink distances with the atomic structure of mLST8 (new Fig. 5b) indicates that the mLST8 β -propeller is closed or at least nearly so.

What happens when LST8 is expressed in E.coli or by in vitro translation? If LST8 is expressed in insect cell expression systems and presented to CCT does it bind? I would like to see more focus on this issue.

The question of the folded state of mLST8 when it is delivered to CCT is an important one but appears beyond the scope of this study. Purifying mLST8 in the absence of CCT or mTOR is a challenge because mLST8 is not stable in the absence of binding partners. Also, the

in vitro binding and folding studies the reviewer suggests may not reflect well what is going on *in vivo*.

The nascent chain squiggles in Figure 6B are highly misleading in my opinion. The WD40 propellers must be significantly folded in order to interact with CCT specifically and with high affinity.

We resorted to squiggles in Figure 6b because we don't know the folded state of mLST8 prior to binding CCT. The hand off may be co-translational with the ribosome (as we point out on p. 21) or other chaperone systems may be involved. We just don't know at this point. We have replaced the squiggles in the figure with more globular "balls of yarn" shapes that may better reflect reality as the reviewer suggests, and we replace the word "unfolded" with "partially folded" in the discussion of Figure 7 on p. 20.

2. The introduction and discussion or the TOR-CCT interaction is shallow. There is a major omission in not discussing the work of Mike Hall's lab (PNAS 1996) which showed genetic interaction between CCT6 and the TOR pathway in yeast; especially since this new work shows that CCT6 interacts with LST8.

We thank the reviewer for pointing out the omission of the paper from the Hall lab. We have included a reference to it in the introduction (p. 4). We also include a statement in the discussion that the folding of mLST8 and Raptor by CCT "affords a possible explanation for the genetic links between CCT and yeast TOR observed previously" (p. 18). These changes succinctly address the concern while keeping the Introduction and Discussion concise.

3. The negative result for PhLPI interaction with the CCT-LST8 system is good to know but opens up other questions. In particular what else is bound to the CCT-LST8 complex purified from the HEK293T cells (Supp fig S7)? The affinity purified material looks clean for CCT subunits; what is the occupancy by LST8? A good analysis would be to subject the CCT-LST8 complex to MALDI-MS analysis to ask what other proteins are bound in addition to LST8? The data may already exist from the X-link mass spec experiment.

We have analyzed the mass spec data of the purified mLST8-CCT samples for other possible binding partners as the reviewer suggested, which identified mTOR and several Hsp70 isoforms as well as a list of common contaminants. We were not surprised by the presence of mTOR since mLST8 was being purified in the last affinity step and mTOR-mLST8 complex has been purified, crystallized and the structure determined (Yang et al. 2013 Nature 497, 217-223). The Hsp70 could be interacting with mLST8 or CCT and contribute to mLST8 folding, but it is also a common contaminant that binds non-specifically to the StrepTactin column, so extensive future investigations would be needed to determine the role, if any, that Hsp70 might play in mLST8 folding.

We also used the mass spec data to determine the relative amount of mLST8 and CCT subunits in the sample by spectral counting, which yielded a three-fold excess of mLST8 to CCT. This value makes sense given that the second affinity step isolated mLST8-containing complexes. The excess of mLST8 increases the likelihood that the CCT particles contain mLST8. Moreover, in the cryo-EM image analysis, we sorted particles for a mass in the center

of the cavity (see Methods p. 29) to ensure that the reconstruction contained only mLST8-CCT particles.

4. A schematic summary of all the transfection data would be useful. Or better labelled gels with sketches above. It is hard work going through the individual gels (which I have done). This data is important though in ruling out what's in and what's out with respect to the CCT-TOR component interactions.

To make the data easier to follow, we have modified the scheme in Figure 2a to help the reader better visualize the workflow of the CCT and PhLP1 knockdown experiments. We have also added labels to each graph in Figure 2 and Figure S3 so the reader can interpret the figures more easily.

Reviewer #3 (Remarks to the Author):

General comments:

The paper is well written and the results provided should be of general interest to the field. The major claim of the paper is that CCT has an essential role in mTOR complex assembly. The Authors have used a number of experimental techniques to demonstrate that this is indeed the case. They have also isolated a late folding intermediate of mLST8 bound to CCT. Surprisingly the binding site for mLST8 was found to be deep within the folding chamber interacting with the N/C-termini of the equatorial domains rather than the apical domains. Cross-linking was used to confirm this binding site. This is an interesting result for the Group 2 chaperonins but I think some more discussion is need on what this means in relation to the group 2 and group 1 chaperonins. I think the paper is worth publishing and have made some specific comments below.

Specific comments:

Abstract:

Sentence 6: Please can the resolution claim be adjusted from 3.97 Angstroms to 4 Angstroms. The FSC resolution estimation is not accurate enough to go to the second decimal place.

This is a good point. The FSC resolution estimate is good enough to report the first decimal place, so we will report 4.0 Angstroms throughout the manuscript.

Final sentence: "which is radically different from the rest of the chaperonins" This reviewer was under the impression that most of the eukaryotic group 2 chaperonins have sequential rather concerted ATP binding/hydrolysis mechanisms.

This statement was not essential to the point of the abstract, so we have removed it.

Results:

Page 7, Paragraph 2: *the half life of the interaction seems very slow for both mLST8 and Raptor is this similar to other known CCT substrates.*

We agree with the reviewer that this release rate is slow, which probably allowed us to successfully purify the mLST8-CCT complex. Data on rates of substrate release from CCT are sparse, but a half-life of 7 min. has been recently reported for ATP-dependent release of folded actin from CCT after dilution from denaturant *in vitro* (Balchin et al. 2018 *Cell* 174, 1507-1521). We have added a statement to reflect this observation on p. 7.

Page 8: It would be interesting to know what happened to the unfolded mLST8 and Raptor in the CCT knockdown cells?

Agreed. Our results suggest that they are being degraded in the absence of CCT because protein expression of the mTORC1 subunits decreases (Fig. 2b) while mRNA expression does not (Supplemental Fig. 2a). The mechanism by which they are targeted for degradation is a topic of future work that we don't believe is essential to the point of the current manuscript.

Page 11 Paragraph 2: What criteria did the authors use to select the 3D class that contained 452,000 particles from 1,197,358. Did the Authors look at the remaining classes?

These criteria are properly explained in the Methods section (pp. 28-29). We carried out an initial 2D classification, and the selected particles were used for an initial 3D classification. Here, we kept the particles that generated volumes with a well-defined CCT structure. With these particles, a second 3D classification was carried out in which we focused on the mass in the interior of the chaperonin cavity, selecting those particles gives rise to a well-defined mLST8 molecule.

Page 11, Paragraph 2: from the local resolution estimation it looks like the intermediate and apical domains are not as well resolved as the equatorial domains. Did the Authors try any further classification to see if a more isotropic reconstructions can be generated from a subset of the 452,000 particles?

This is a good suggestion that we tried, but unfortunately the resolution did not improve. We tried to classify these 452,000 particles, but as you can see in the figure below, in all the classes generated there was poor definition in some of the apical domains (see asterisks in the figure below)

Page 12, Final Paragraph: The mLST8 seems to fit well, however I think it would help the reader if the density was extracted for mLST8 and shown with the coordinates as it is difficult to truly assess the quality of fit when shown with the rest of the CCT structure. This could be a supplementary figure.

This is an excellent suggestion that we have included in a new Figure 4c. The figure clearly shows the close fit of the mLST8 atomic structure into the cryo-EM density extracted from between the rings.

Page 13, First Paragraph: Was poly lysine used for the control CCT grids? Or did the authors not have a preferred orientation in the absence of substrate.

Without any treatment, CCT always shows a preferred orientation (the end-on view). Poly lysine has helped us to overcome this problem for both samples (apo CCT and the CCT-mLST8 complex). We added a statement on p. 27 of the Methods to clarify that grid preparation was the same for both complexes.

Page 13, The resolution of the control is much lower than that of the substrate bound complex? There are 100,000 less particles but I am not sure that explains the difference. It would be good for the authors to show the angular coverage they get from the CCT only structure and offer a potential reason for this.

We incorrectly reported in the main text the number of particles in the best classes as 348,446 when the final number was 139,819 (p. 12). The methods section correctly reported the number of particles used. The actual difference in the number of particles between the two

samples was 300,000, which we believe explains the difference in resolution. Regarding the angular coverage, the figure below shows the angular coverage for the apo-CCT structure, which basically is very similar to that already shown for the CCT-mLST8 complex.

Discussion:

Page 19, Paragraph 1: If the main binding site for the mLST8 is located at the N/C termini what is the role of the apical domains in binding/folding of mLST8? In an ideal world the authors would confirm the C/N termini as the primary binding sites by deletion studies. This has been done for GroEL, and has been shown to have an effect on folding efficiency of some substrates. I understand that making the mutants is a large undertaking and probably out of the scope of this study. However, I think some more discussion is needed and some comparisons to the group I chaperonins may help (Weaver and Rye, 2014).

We agree with the reviewer that deleting the N/C termini of specific subunits would help to clarify the role of these as substrate primary binding sites, but also agree that doing this mutation study is a large undertaking that we plan to do in the future. At the reviewer's suggestion, we have expanded our discussion of the interactions of substrates with group I chaperonins, pointing out that most bind the apical domains but that interactions have been observed with the termini (p. 18).

Methods:

Page 32, final paragraph: for the Data Acquisition on mLST8-CCT the Authors say they collected 2753 movies but in the next page for the image processing they then say that they motion corrected 5576 movies. Please can this be clarified.

This was a mistake. 5576 movies should have been stated in both places. We have corrected the error (p. 27)

Page 33, Second paragraph: "The MotionCorr2 output was subjected to CTF correction using CTFFIND4" change to "The MotionCorr2 output was subjected to CTF determination using CTFFIND4"

We made the change (p. 28).

Page 33, Second paragraph: Which initial model was used and why was it selected. I think some more detail may be useful.

As described in the Methods, two programs were used (EMAN and RANSAC) to generate an initial volume, and with both we obtained a similar result, low-resolution cylinders with similar dimensions to CCT. This information has been included in the manuscript (p. 28).

Page 34, first paragraph: the local resolution estimation is not in Table S1. Could the authors provide the estimated local resolution (mean or median value) and the histogram for the resolution bins.

The mean resolution is 5.67 Å and the median 4.6 Å. These values have been included in Table S1. A histogram with the resolution reached for every voxel has been included as a new Fig. S4g.

Page 34, 35, model building: What restraints did the authors use in phenix during real space refinement? As at this resolution refining non-bulky hydrophobic side chains could be difficult especially in the apical and intermediate domain regions.

The restraints used in phenix real-space refinement were both the standard (bond, angle, planarity, chirality, dihedral and nonbonded repulsion), and some extra restraints (Ramachandran plot, C-beta deviations, rotamer and secondary structure). A local grid search-based fit was included in the refinement strategy to fix side-chain outliers (rotamers or poor map fitting). This information has been added on p. 30.

Reviewer: Daniel Clare

Reviewer #4 (Remarks to the Author):

Cuéllar et al. provide exciting data on the folding of subunits of the mTOR complexes by the chaperonin CCT. Specifically, they obtained a high-resolution (~4 Å) cryoEM structure of the mLST8 subunit trapped inside the CCT folding chamber. In contrast to previous structures of CCT substrates, mLST8 is observed to be positioned deep inside the folding chamber, between the two rings of CCT.

The authors first show, using co-IP, that both mLST8 and a second mTOR complex subunit, Raptor, show strong interactions with CCT, lending support to the hypothesis that the beta-

propellers of the two proteins are folded by CCT. This conclusion is further supported by cell viability assays, highlighting the role of CCT for mTOR complex assembly.

The cryoEM structure of mLST8 bound to CCT allowed a precise positioning of the substrate in the folding chamber: between the two rings and asymmetrically located towards subunits 3,6 and 8 of CCT, i.e. in the "CCT6 hemisphere" that exhibits weaker ATP binding and hydrolysis. Additionally, the authors were able to observe ATP occupancies in the structure of the mLST8-CCT complex. In addition to EM, the authors also performed crosslinking/mass spectrometry (XL-MS) experiments that provided some crosslinks on mLST8 itself and between mLST8 and CCT subunits, although overall I do not consider the XL data very helpful in support of the EM data (see specific comments below).

Together, Valpuesta et al. present very interesting results on the folding of mTORC subunits by CCT, providing new insights into the chaperonin's role in TOR pathways. The data will certainly help to better understand of the different ways that CCT can fold its substrates, as in this case a different positioning compared to other beta-propeller containing proteins is observed and a connection to ATP binding and hydrolysis can be made.

Specific comments related to XL-MS data:

The evidence from XL-MS as presented on pages 14-15 and in Figure 4 is weak. Obtaining a sufficient number of crosslinks on a substrate trapped in the folding chamber is certainly challenging as a result of the timescale of folding and the general preference of forming crosslinks on exposed residues, most probably on the outer surface of CCT. The crosslinking results are certainly consistent with the positioning of mLST8 as derived from cryo-EM, but would fit to many other alternative locations of the substrate equally well because they only connect only a single residue of mLST8 (Lys 215) to several TRiC subunits. So, even if the substrate would be flipped to the other side of the folding chamber and predominantly interact with subunits of the other hemisphere, then the distance restraints would likely be fulfilled, at least judging from Fig. 4c. In this context, what does "high-quality crosslink" mean in the figure legend (page 39)?

In the time since the manuscript was first submitted, we have reanalyzed our XL-MS data because we had discovered an error in the mass we were using in pLink for the D2 labeled DSS crosslinks. (As indicated in the Methods p. 31, we used a 50:50 mix of H2/D2 labeled DSS for the crosslinking.) In addition, we included a list of 293 common contaminants provided by pLink in the search database to decrease false positive identifications. These modifications improved the quality of the crosslink data. For example, of the 196 unique links within CCT, only 21 exceeded the 32 Å distance constraint and all 48 links in the structurally stable CCT equatorial domains were within the distance constraint. This improved search identified 2 additional intermolecular K215 mLST8-CCT links, giving a total of 5 intermolecular K215 mLST8-CCT links that orient mLST8 in the cryo-EM density more clearly. The distance constraints strongly favor an orientation with mLST8 K215 on the right toward the center of the folding cavity as shown in new Figure 5a. In this orientation, the docking of mLST8 in the cryo-EM density is also excellent (new Figure 4c). Thus, the evidence for the proposed orientation of

mLST8 is stronger than in the original manuscript and we would like to retain the crosslinking data. We have rewritten the paragraph on p. 14 to reflect these differences and we dropped the term “high-quality crosslink”.

It is also somewhat surprising that most of the crosslinks observed on mLST8 itself are formed across the propeller (Fig. 4d), bridging relatively large distances, compared to the linking of more adjacent Lys residues (that are involved in crosslinks across the molecule). Can this be explained by the specific orientation of the lysines on the beta-sheets?

The crosslinking reanalysis lost 1 intralink in mLST8 while retaining the other 8 intralinks. Of these, 2 are across the β -propeller and beyond the distance constraint while 6 are on the same side of the β -propeller and within the distance constraint (Figure 5b), so most of the intralinks are actually not across the β -propeller. These intralinks support the observation from the cryo-EM density that mLST8 is in a near-native structure (Figure 4c), especially considering that there are 8 other possible links beyond the distance limit that we did not see. Two lysines are not involved in crosslinks, K158 and K213, but their orientation does not preclude crosslinks, suggesting that they are unreactive or their peptides do not ionize well. We have modified the paragraph on pp. 14-15 to reflect these observations.

In case the authors decide to keep the XL data in the manuscript, the relevance for making conclusions about the positioning/orientation of mLST8 need to be toned down a bit and more information than the selected few crosslinks listed in Table S3 needs to be provided. All crosslinks, also those on CCT, need to be listed together with essential data such as the exact peptide sequences that were identified, confidence scores, mass errors etc.

We have rewritten the XL-MS section (pp. 13-15) to include the revised data and to ensure a stronger justification of the conclusions by the data. We have changed Supplementary Table 3 to include the crosslinks within and between the highly structured equatorial domains of the CCT subunits. As mentioned, all of the 48 links in this region were within the 32 Å distance constraint, supporting the accuracy of the structural model and the quality of the crosslinking data. The pLink output, which contains all the crosslink information the reviewer requests, is available in the Source Data File, and the raw mass spec data are available on the Chorus project repository, project:1567, experiment: 3379.

Reviewed by Alexander Leitner, ETH Zurich, Switzerland